# Structural basis for the recognition of complex-type N-glycans by Endoglycosidase S

Beatriz Trastoy[1], Erik Klontz[2,3], Jared Orwenyo[4], Alberto Marina[1], Lai-Xi Wang[4], Eric J. Sundberg[2,3,5] & Marcelo E. Guerin[1,6,7,8]

Endoglycosidase S (EndoS) is a bacterial endo-β-N-acetylglucosaminidase that specifically catalyzes the hydrolysis of the β-1,4 linkage between the first two N-acetylglucosamine residues of the biantennary complex-type N-linked glycans of IgG Fc regions. It is used for the chemoenzymatic synthesis of homogeneously glycosylated antibodies with improved therapeutic properties, but the molecular basis for its substrate specificity is unknown. Here, we report the crystal structure of the full-length EndoS in complex with its oligosaccharide G2 product. The glycoside hydrolase domain contains two well-defined asymmetric grooves that accommodate the complex-type N-linked glycan antennae near the active site. Several loops shape the glycan binding site, thereby governing the strict substrate specificity of EndoS. Comparing the arrangement of these loops within EndoS and related endoglycosidases, reveals distinct-binding site architectures that correlate with the respective glycan specificities, providing a basis for the bioengineering of endoglycosidases to tailor the chemoenzymatic synthesis of monoclonal antibodies.

[1] Structural Biology Unit, CIC bioGUNE, Bizkaia Technology Park, 48160 Derio, Spain. [2] Institute of Human Virology, University of Maryland School of Medicine, Baltimore, MD 21201, USA. [3] Department of Medicine, University of Maryland School of Medicine, Baltimore, MD 21201, USA. [4] Department of Chemistry and Biochemistry, University of Maryland, College Park, MD 20742, USA. [5] Department of Microbiology and Immunology, University of Maryland School of Medicine, Baltimore, MD 21201, USA. [6] Unidad de Biofísica, Centro Mixto Consejo Superior de Investigaciones Científicas—Universidad del País Vasco/Euskal Herriko Unibertsitatea (CSIC,UPV/EHU), Barrio Sarriena s/n Leioa, Bizkaia 48940, Spain. [7] Departamento de Bioquímica, Universidad del País Vasco, Leioa 48940, Spain. [8] IKERBASQUE, Basque Foundation for Science, 48013 Bilbao, Spain. These authors contributed equally: Beatriz Trastoy, Erik Klontz. Correspondence and requests for materials should be addressed to E.J.S. (email: ESundberg@som.umaryland.edu) or to M.E.G. (email: mrcguerin@cicbiogune.es)

Therapeutic immunoglobulin G (IgG) antibodies are a prominent and expanding class of drugs used for the treatment of several human disorders including cancer, autoimmunity, and infectious diseases[1–3]. IgG antibodies are glycoproteins containing a conserved N-linked glycosylation site at residue Asn297 on each of the constant heavy chain 2 (CH2) domains of the fragment crystallizable (Fc) region (Fig. 1)[4]. The presence of this N-linked glycan is critical for IgG function[5,6], contributing both to Fc γ receptor binding and activation of the complement pathway[7,8]. The precise chemical structure of the N-linked glycan modulates the effector functions mediated by the Fc domain[9]. IgG antibodies including those produced for clinical use typically exist as mixtures of more than 20 glycoforms, which significantly impacts their efficacies, stabilities and the effector functions[10,11]. To better control their therapeutic properties, the chemoenzymatic synthesis of homogeneously N-glycosylated antibodies has been developed[12–14].

Endoglycosidase S (EndoS) secreted from *Streptococcus pyogenes* is a 108 kDa enzyme that specifically catalyzes the hydrolysis of the β-1,4 linkage between the first two *N*-acetylglucosamine residues of the complex-type N-linked glycan located on N297 of the Fc region of IgG antibodies (Fig. 1)[15,16]. This structural modification ablates the effector functions of the host IgG antibodies, markedly contributing to immune evasion by this bacterium[17] because (i) EndoS deglycosylates only IgG glycoforms and no other glycoproteins, and (ii) EndoS glycosynthase variants efficiently transfer predefined complex-type N-linked glycans to intact IgG, this endoglycosidase plays a central role in glycoengineering strategies to develop IgG antibodies with improved therapeutic potential[12–16]. Recently, we have described the X-ray crystal structure of a truncated version of EndoS (98–995) in its unliganded form[18]. However, the molecular mechanism by which EndoS specifically recognizes biantennary complex-type glycans linked to N297 of IgG remains unclear, prohibiting the full exploitation of this enzyme in therapeutic antibody engineering.

Here X-ray crystallography, small-angle X-ray scattering (SAXS), site-directed mutagenesis, enzymatic activity, and computational methods are used to define the molecular basis of substrate specificity of EndoS, as well as that of other GH18 endoglycosidase family members.

## Results

**Overall structure of full-length EndoS$_{D233A/E235L}$-G2 complex.** For our structural studies, we used a catalytically inactive version of EndoS, in which the residues D233 and E235 are mutated to alanine and leucine, respectively (EndoS$_{D233A/E235L}$, see below for further details). The crystal structure of the full-length catalytically inactive EndoS$_{D233A/E235L}$ (residues 37–995; residues 1–36 correspond to the signal peptide) in complex with G2 product was solved by molecular replacement methods (EndoS$_{D233A/E235L}$-G2 thereafter; Fig. 2; Supplementary Figs. 1 and 2;

Supplementary Table 1 and Methods section)[18]. EndoS$_{D233A/E235L}$ crystallized in the $P2_1$ space group, with one molecule in the asymmetric unit and diffracted to a maximum resolution of 2.9 Å (Supplementary Table 1). The full-length EndoS comprises six different domains from the N- to the C-terminus: (i) the N-terminal domain (residues 37–97) determined by the EndoS$_{D233A/E235L}$-G2 crystal structure adopts a three-helix bundle domain (N-3HB) that is connected to the previously reported (ii) glycosidase domain (residues 113–445) by a proline-rich, 15 residue-long loop (residues 98–112; Fig. 2); (iii) a leucine-rich repeat domain (residues 446–631); (iv) a hybrid Ig domain (residues 632–764) that comprises two subdomains that are topologically intertwined, a typical Ig subdomain structurally similar to the interleukin-4 receptor (PDB code 1IAR; Z-score = 5.2) with an insertion of a smaller subdomain between the second and third β-strands; (v) a carbohydrate binding module (residues 765–923), and (vi) a C-terminal three-helix bundle domain (C-3HB; residues 924–995)[18]. The structural comparison between the full-length EndoS$_{D233A/E235L}$-G2 and the truncated unliganded version ΔN-3HB-EndoS suggests an important contribution of the N-3HB domain to stabilize the GH domain and generate a completely competent glycan binding site (Supplementary Fig. 3a–d). Supporting this notion, the calculated-buried surface area between the N-3HB and GH domains is 464 Å$^2$ [19]. Specifically, Q91 and E94 at the end of α3 form hydrogen bonds with the main chain of Y157 of loop 2 and K162 of α4, respectively. In addition, S57 at the end of α2 forms hydrogen bonds with D156 of loop 2. This interface region is further stabilized by hydrophobic interactions mediated by F64, L56, L95, Y157, and L159 (Supplementary Fig. 3e). To study the thermostability of the full-length EndoS$_{D233A/E235L}$ and ΔN-3HB-EndoS, we performed differential scanning fluorimetry (DSF). ΔN-3HB-EndoS shows two well-separated unfolding transition states with melting temperature ($T_m$) of 45 and 51 °C, whereas the full-length EndoS$_{D233A/E235L}$ only displays one transition at 51 °C (Supplementary Fig. 3f), consistent with the notion that the N-3HB domain contributes to stabilize EndoS.

The N-3HB was previously suggested to be an oligomerization domain[18]. However, EndoS$_{D233A/E235L}$-G2 crystallized as a monomer, and this monomeric state was confirmed to occur in solution, both in the presence and absence of the oligosaccharide G2 product by SAXS (Supplementary Fig. 4). The radius of gyration ($R_g$) values obtained for EndoS$_{D233A/E235L}$ in presence (43.9 Å) and absence (43.2 Å) of the G2 product revealed a slight reduction of the $R_g$ value of about 0.7 Å. In addition, we observed a similar fit of the SAXS data in the presence and absence of the G2 product in the solution-scattering profile calculated from the crystal structure of EndoS$_{D233A/E235L}$-G2 product complex, suggesting that the overall shape of the enzyme remains unchanged (Supplementary Table 2; Supplementary Fig. 4 and Methods section). The N-3HB domain is structurally similar to the Staphylococcal protein A (SpA) C domain (PDB ID code

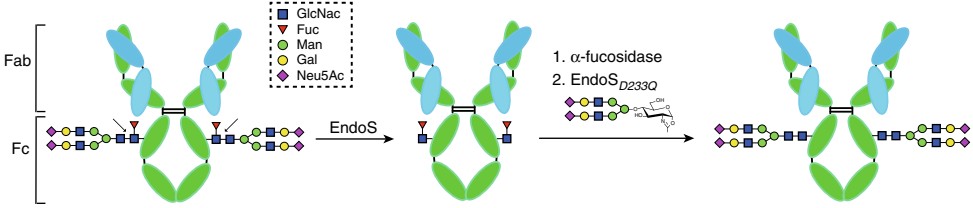

**Fig. 1** Schematic representation of EndoS hydrolytic activity and glycosynthase activity of EndoS mutant. EndoS specifically hydrolyzes the β-1,4 linkage between the first two *N*-acetylglucosamine residues of the complex-type N-linked glycan located on Asn297 of the Fc region of IgG antibodies. The N-linked glycan is represented exposed on the exterior of the Fc domain to facilitate its visualization. EndoS mutant (EndoS$_{D233Q}$) efficiently transfers fucosylated and afucosylated biantennary complex-type N-linked oligosaccharide from complex-type sugar oxazoline as a donor substrate

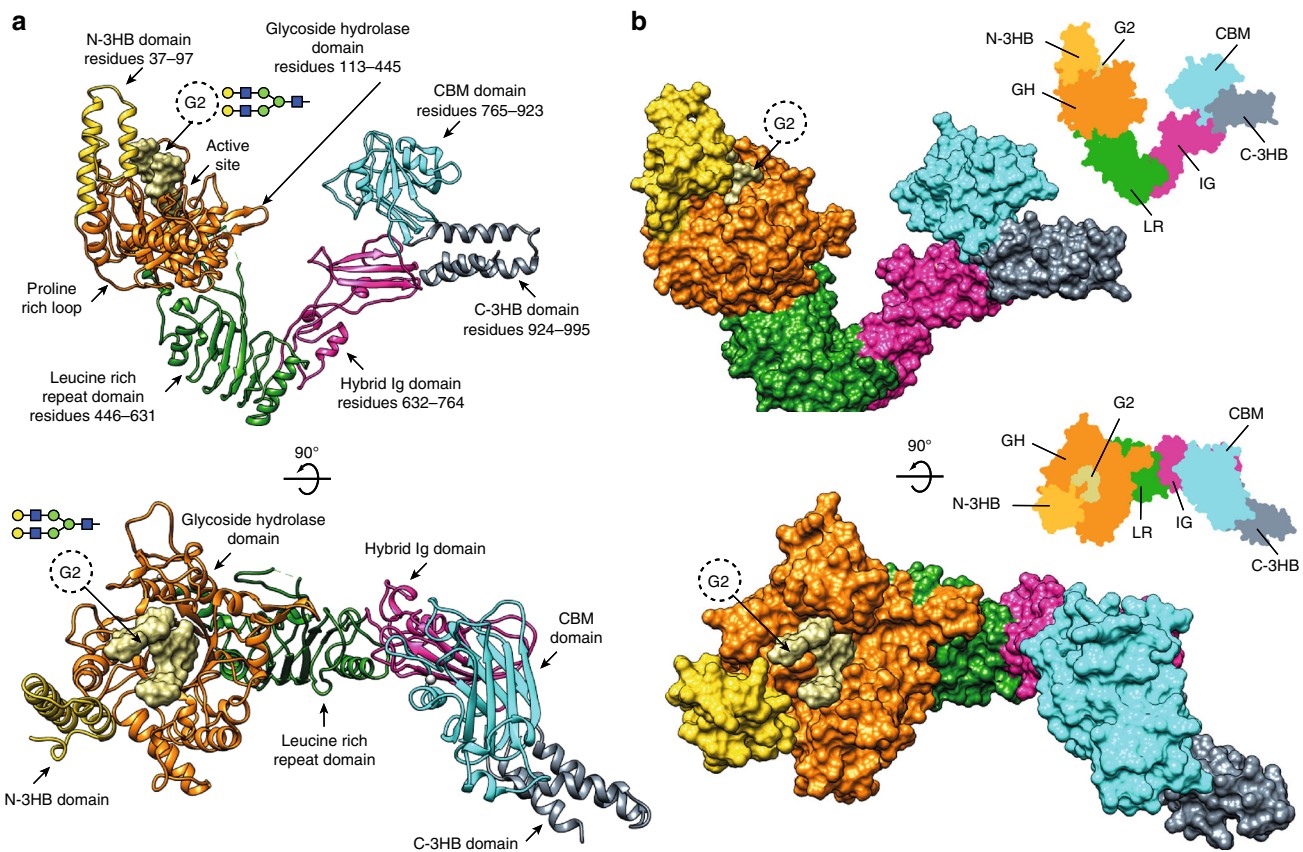

**Fig. 2** Overall structure of the EndoS$_{D233A/E235L}$-G2 complex. **a** Cartoon representation showing the general fold and secondary structure organization of EndoS$_{D233A/E235L}$, including the N-3HB (yellow), glycoside hydrolase (GH; orange), leucine-rich repeat (green), hybrid IgG (magenta), carbohydrate binding module (cyan), and C-3HB (gray) domains. The G2 product is shown in light brown. **b** Surface representation of the EndoS$_{D233A/E235L}$-G2 complex showing the location of the G2 product binding site and catalytic site

4ZNC; Z-score = 6.6), a 42-kDa protein that contains five highly homologous extracellular Ig-binding domains in tandem, designated domains are E, D, A, B, and C. The SpA C domain binds between the CH2 and CH3 domain of the Fc region of IgG[20]. In addition, the SpA D domain also binds to the human Fab-heavy chain of the V$_H$3 family, assisting *Staphylococcus aureus* in evading the immune system[20,21].

**The G2 product binding site**. The EndoS glycosidase domain adopts a $(\beta/\alpha)_8$-barrel topology with a long cavity that runs parallel to the protein surface in which one molecule of the G2 glycan product is unambiguously identified in the crystal structure (Figs. 2–4). Specifically, the G2 glycan product binding site is located in the central region of the β-barrel and is flanked by α2 and α3 helices from the N-3HB domain, as well as the connecting loops β1–β2 (loop 1; residues 120–145), β2–α4 (loop 2; residues 151–158), β3–α5 (loop 3; residues 185–206), β4–α6 (loop 4; residues 235–247), β5–α7 (loop 5; residues 281–289), β6–α8 (loop 6; residues 304–306), β9–α10 (loop 7; residues 347–380), β10–α11 (loop 8; residues 403–413), and α11–α12 (loop 9; residues 420–434).

The reducing end of the core Manβ1–4GlcNAc disaccharide is located at the end of the long cavity flanked by loops 4, 5, 6, and 7, and several residues from the β-barrel core (Fig. 3a, b). Two well-defined asymmetric grooves accommodate each of the complex-type N-linked glycan antennae: the Galβ1–4GlcNAcβ1–2Manα1–6 and Galβ1–4GlcNAcβ1–2Manα1–3 arms occupy grooves 1 and 2, respectively, both attached to the disaccharide Manβ1–4GlcNAc of the G2 product (Fig. 3a–c).

Specifically, the O6 atom of the first GlcNAc (−1) residue interacts with the side chains of E349, N356, and W358, whereas O1 interacts with the side chains of Q303 and Y305 (Fig. 3d). The O2 atom of the Man (−2) residue interacts with the side chains of E349 and Y402, while its O4 atom makes a hydrogen bond with the indole nitrogen of W153 and F150 stacks against the sugar ring of Man (−2). W153 is positioned in such a way as to engage the entire G2 trimannose core including the central Man (−2), the α(1–6)-linked Man (−3) and the α(1–3)-linked Man (−7). The O3 atom of the Man (−3) residue interacts with the side chains of R186 and D237, whereas the O6 atom makes a hydrogen bond with the main chain of H151. In addition, the O3 atom of the Man (−7) residue interacts with R119 and E350, while the O4 and O6 atoms make a hydrogen bond with the main chains of E350 and A352, respectively. W121 also stacks against the sugar ring of the Man (−7) residue. The terminal GlcNAc (−4 and −8) and Gal (−5 and −9) residues of each arm adopt two alternative conformations into the crystal structure. In one state, the GlcNAc (−4 and −8) and Gal (−5 and −9) residues protrude away from grooves 1 and 2, which may reflect the release of the G2 product from the active site (Fig. 4a); in the other state, these carbohydrates reside within their corresponding grooves, likely reflecting the binding mode of the G2 substrate (Fig. 4c).

Although we were unable to co-crystallize EndoS in the complex with the S2G2 substrate (Fig. 5a), the three-dimensional structure suggests the possible binding mode for the first GlcNAc (+1) and the last Neu5Ac (−6 and −10) residues (Fig. 4b, d). Molecular docking calculations placed the GlcNAc (+1) residue

within a region located at the end of the long cavity comprising β6 and loops 4, 5, 6, and 7, and the last two Neu5Ac (−6 and −10) residues of the S2G2 substrate extending beyond the glycan binding grooves 1 and 2, respectively (Fig. 4b, d). The O6 atom of GlcNAc (+1) makes a hydrogen bond with the side chain of T281, whereas the O1 atom and the carbonyl oxygen of the *N*-acetamido group of sugar interacts with the main chain of Q303 and Y305, respectively. The GlcNAc (+1) is also stabilized by hydrophobic interactions mediated by Y305 and W358 (Fig. 5b). EndoS belongs to family GH18, for which a substrate-assisted mechanism, with retention of the anomeric configuration, has been proposed[22–25]. During the first step, the binding of the substrate generates a distortion of GlcNAc (−1), preceding the transfer of a proton from a protonated carboxylic acid residue to the anomeric oxygen, and the nucleophlic attack at the anomeric center by the carbonyl oxygen of the *N*-acetamido group to result in the formation of an oxazolinium intermediate[22–26]. A second carboxylate residue is thought to orient and enhance the nucleophilicity of the acetamido group that attacks the anomeric center by formation of a hydrogen bond[25]. During the second

step, the general acid residue in the first step is proposed to deprotonate an incoming water. This water molecule promotes the departure of the 2-acetamido group, releasing the sugar hemiacetal product with overall retention of stereochemistry[22]. Critical residues are preserved in EndoS, strongly supporting a common catalytic mechanism (Fig. 5c and Supplementary Fig. 5). In that context, E235 base/base, whereas D233 stabilizes the intermediate in a substrate-assisted hydrolysis mechanism, in which the carbonyl group of the C2-acetamido of GlcNAc (−1) acts as the nucleophile (Fig. 5b)[22]. For that reason, we have replaced both D233 and E235 residues by alanine and leucine, respectively, in order to obtain a catalytically inactive enzyme (EndoS$_{D233A/E235L}$) for further use in our structural studies. In addition, D231 provides a negative charge that keeps D233-E235 protonated, whereas Y402 stabilize the transition state[27].

To further investigate the mechanism of substrate specificity of EndoS at the molecular level, we mutated the loops that decorate the β-barrel core of EndoS and contact the G2 product glycan in our crystal structure, and studied their ability to process the N-linked glycan on Rituximab, a chimeric monoclonal antibody

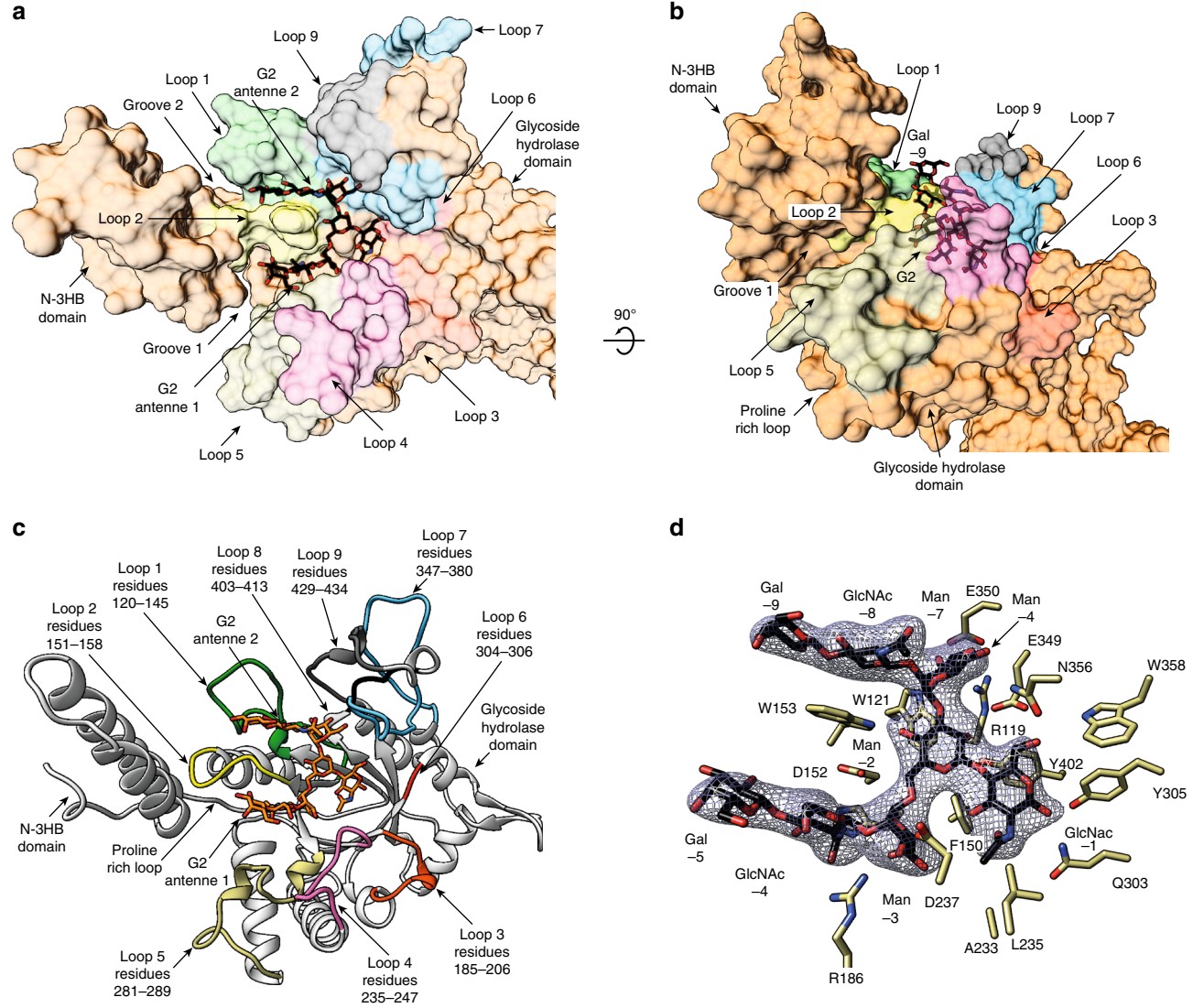

**Fig. 3** The G2 product binding site of EndoS. **a, b** Surface representation of EndoS$_{D233A/E235L}$ showing the loops surrounding the active site of the glycosidase domain (orange) and the location of the two well-defined asymmetric grooves that accommodate each of the G2 product complex-type N-linked glycan antennas (black). **c** Cartoon representation showing the loops that decorate the G2 product (in orange) binding site. **d** Key residues of EndoS interacting with G2 product are colored light brown. The corresponding electron density of G2 product is shown at 1.0 σ r.m.s. deviation

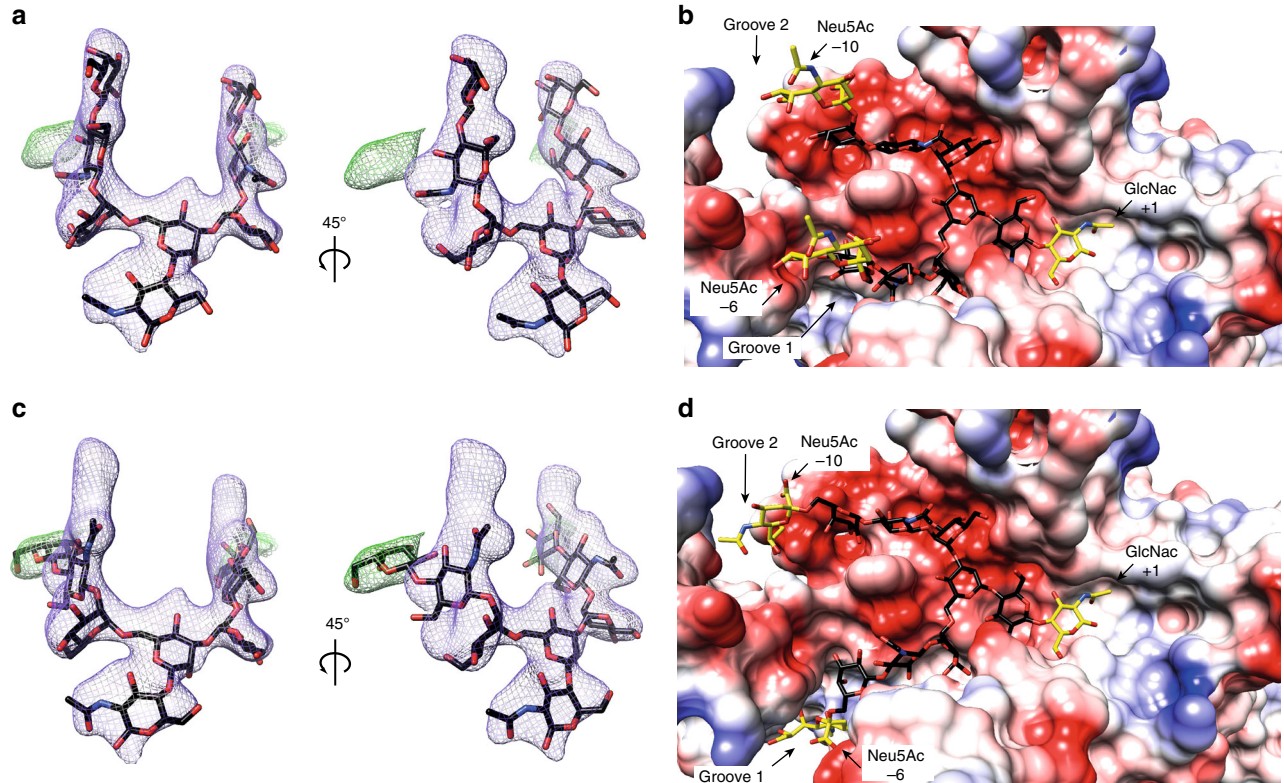

**Fig. 4** Electron density map showing the two alternative conformations of G2 product. **a** Two views of the final electron density maps ($2mF_o–DF_c$ contoured at 1σ (purple) and $mF_o–DF_c$ at 1σ (green)) corresponding to the conformation of G2 product outside the grooves 1 and 2. **b** Electrostatic surface representation of the EndoS$_{D233AE235L}$-S2G2 substrate complex model showing the S2G2 substrate outside the groove. The first GlcNAc (−1) and the last Neu5Ac (+6 and +10) residues were modeled and are shown in yellow (Methods section). **c** Two views of the final electron density maps ($2mF_o–DF_c$ contoured at 1σ (purple) and $mF_o–DF_c$ at 1σ (green)) corresponding to the conformation of G2 product inside the grooves 1 and 2. **d** Electrostatic surface representation of the EndoS$_{D233AE235L}$-S2G2 substrate complex model showing the S2G2 substrate inside the groove. The first GlcNAc (−1) and the last Neu5Ac (+6 and +10) residues were modeled and are shown in yellow (Methods section)

bearing a human IgG1 Fc region and approved for the treatment of B-cell lymphoma (Fig. 6). Specifically, we made alanine mutations of the key residues in loop 1 (R119, E130, and K133), loop 2 (W153), loop 3 (R186 and N193), loop 4 (D237 and K241), loop 6 (Q303 and Y305), and loop 7 (S346, E349, E350, and E356; Fig. 3c, d and Fig. 6a). As depicted in Fig. 6b, mutations in loops 1, 6, and 7 completely abolished the hydrolytic activity of the enzyme. Loop 6 mediated the interaction of EndoS with GlcNAc (+1), while loop 1 and 7 did so with the antenna 2 of the complex-type N-linked glycan. Mutations in loop 3 significantly decreased the hydrolytic activity of the enzyme, whereas the mutations in the solvent exposed-loop 4 variant were less impactful. Both of these loops were involved in the recognition of antenna 1 of the complex-type N-linked glycan (Fig. 6b). Collectively, the mutational analysis of the EndoS loops that contact the glycan indicated that of the two antennae of G2, interactions with antenna 2 (loops 1 and 7) were critical for glycan recognition, while those with antenna 1 (loops 3 and 4) were nearly dispensable. Finally, the replacement of W153, located in loop 2, with alanine significantly reduced the hydrolytic activity of EndoS, consistent with the position of its side chain that bisected grooves 1 and 2 (Figs. 3c, d and 4a, b; Supplementary Fig. 3c, d). The deletion of the N-3HB domain flanking the extremities of both grooves also resulted in a substantial reduction of the glycoside hydrolase activity and the binding affinity to Rituximab (Fig. 6b)[18]. Altogether, these structural data certainly contributed to define the structural basis for the biantennary complex glycan specificity of EndoS.

## Discussion

To further advance the understanding of EndoS glycan specificity, we performed a structural analysis in the context of the GH18 family of endoglycosidases. A search for structural homologues using the DALI server revealed five endoglycosidases of the GH18 family with significant structural similarity to EndoS: (i) EndoF$_3$ from *Elizabethkingia meningoseptica* (PDB code 1EOM; DALI Z-score of 21.9; r.m.s.d. value of 2.7 Å for 234 aligned residues; 18% identity)[28,29], (ii) EndoT from *Trichoderma reesei* RUT-C30 (PDB code 4AC1; Z-score of 19.0; r.m.s.d. value of 3.1 Å for 242 aligned residues, 12% identity)[30], (iii) EndoH from *Streptomyces plicatus* (PDB code 1C8Y; Z-score of 16.5; r.m.s.d. value of 2.7 Å for 214 aligned residues; 18% identity)[31], (iv) EndoF$_1$ from *E. meningoseptica* (PDB code 2EBN; Z-score of 15.1; r.m.s.d. value of 3.1 Å for 213 aligned residues; 13% identity)[32] and (v) EndoBT from *Bacteroides thetaiotaomicron* VPI-5482 (PDB code 3POH; Z-score of 13.2; r.m.s.d. value of 3.3 Å for 210 aligned residues; 13% identity; Supplementary Fig. 5). A structural comparison of EndoS with the other five members of the GH18 family of endoglycosidases highlights the unique specificity of this enzyme. The glycoside hydrolase domains adopt a (β/α)$_8$ topology, with a series of loops that decorate the β-barrel and build the majority of the carbohydrate binding site, defining substrate specificity (Fig. 7). The crystal structure of EndoF$_3$ was solved in both its unliganded form and in complex with the G2 product, whereas the structures of EndoT, EndoH, EndoF$_1$, and EndoBT were solved in their unliganded forms. EndoF$_3$ hydrolyzes both biantennary and triantennary complex-type N-linked glycans of IgG Fc regions and other glycoproteins[33]. A detailed comparison of

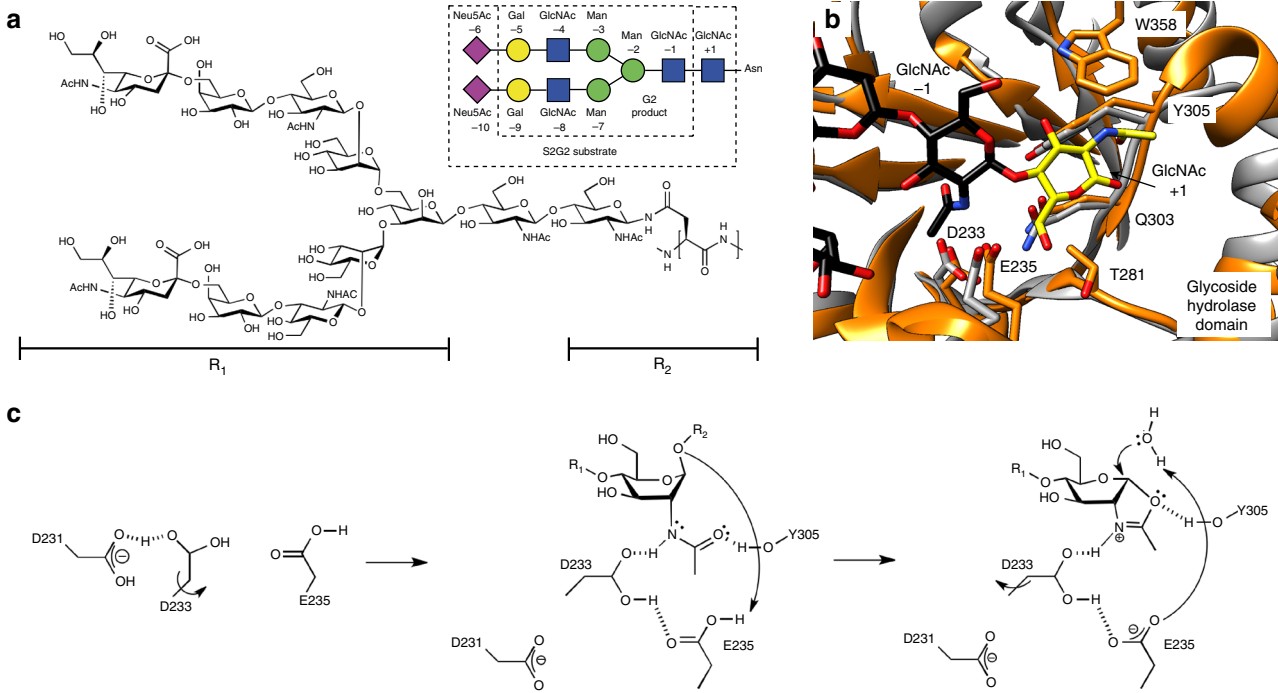

**Fig. 5** The catalytic mechanism of EndoS. **a** Chemical structure of S2G2 substrate. Inset: symbol representation of S2G2 substrate and G2 product. **b** Structural superimposition of the catalytic site of EndoS (orange) and EndoF₃ (gray). EndoS residues are numbered. The GlcNAc (−1) residue was modeled and is shown in yellow. A233 and L235 were replaced by the native aspartic and glutamic acids, respectively. **c** In the resting enzyme, D233 is too far away to interact with E235. In the first step, the binding of the substrate generates a distortion of the GlcNAc (−1) subunit and rotation of D233 toward E235, enabling hydrogen bond interactions between the hydrogen of the acetamido group, D233 and E235. In the second step, the hydrolysis of the oxazolinium ion intermediate leads to protonation of E235 and rotation of D233 to its original position, where it shares a proton with D231

the EndoS_D233A/E235L-G2 and EndoF₃-G2 product complexes clearly explains how EndoF₃ accommodates the same biantennary product, while also accepting triantennary glycans. As depicted in Fig. 8, the G2 product binds to a wide, solvent exposed cavity of ca. 1160 Å³ volume in EndoF₃, whereas the same product is buried deep into a well-defined cavity of ca. 2960 Å³ in EndoS. EndoF₃ and EndoS exhibit a strong resemblance in their catalytic sites. EndoF₃ residues D126, E128, and Y213 lie in equivalent positions as D233, E235, and Y305 in EndoS. Moreover, residues that interact with the innermost part of the G2 product including the GlcNAc (−1), Man (−2), Man (−3), and Man (−7) core are also conserved between the two enzymes. The side chains of Q211 and E245 interact respectively with the O6 atom of GlcNAc (−1) and O2 atom of Man (−2) in EndoF₃ as the equivalent to Q303 and E349 in EndoS. The aromatic rings of F39 and Y472 interact respectively with the GlcNAc (−1), Man (−2), and Man (−3) core, and O2 atom of Man (−2), equivalent to F150 and Y402 in EndoS. The terminal GlcNAc (−4 and −8) and Gal (−5 and −9) residues interact with EndoF₃ and EndoS through a completely different network of hydrogen bonds and hydrophobic interactions. The most important differences are observed in loops 2 and 7 (Figs. 7 and 8). EndoF₃ displays a long loop 2 (residues 41 to 64) including a 1.5 turn α-helix, which is absent in the shorter version of the equivalent loop in EndoS. In EndoF₃, loop 2 only interacts with antenna 1 of the G2 product, whereas the corresponding loop in EndoS clearly interacts with both antennae of the G2 product, bisecting the binding cavity into two grooves. In addition, loop 7 in EndoF₃ is markedly shorter than that observed in EndoS. As depicted in Fig. 8, EndoF₃ exhibits a cavity sufficient to accommodate antenna 3, and the extra antenna cannot be accommodated into the EndoS grooves due to steric hindrance (Methods section). Thus, EndoS contains a particular groove 2, significantly different from EndoF₃, which allows the enzyme to

selectively recognize the biantennary complex-type N-linked oligosaccharides. Altogether, these structural features of EndoF₃ and EndoS certainly explain the unique hydrolytic specificity of each enzyme.

Inspection of the EndoT, EndoH, and EndoF₁ crystal structures, all high-mannose type-specific endoglycosidases, revealed substantial differences in the architecture of the putative oligosaccharide binding cavity when compared to that of EndoS: (i) loop 1 is structurally ordered; (ii) loop 2 adopts a β-hairpin conformation that extends away from the central core of the enzyme, likely involved in the recognition of antenna 1 of the high-mannose-type N-linked glycans[34,35]; and (iii) loop 7 is markedly shorter (Fig. 7). Molecular docking calculations placed a high-mannose-type oligosaccharide into the putative oligosaccharide binding cavity (Fig. 9a–d). The calculated volume of the cavity was ca. 2047, 1865, and 1672 Å³, for EndoT, EndoH, and EndoF₁, respectively. Antenna 1 of the high-mannose-type oligosaccharide makes contacts with residues of loop 2 and 3; antenna 2 interacts with residues of loop 4; and antenna 3 makes contacts with loop 1 and 9 residues. The crystal structure of EndoBT, a putative endoglycosidase of unknown function, was solved in its unliganded form (PDB code 3POH). In contrast to EndoT, EndoH, EndoF₁, and EndoF₃, EndoBT contains and additional carbohydrate binding module domain (Supplementary Fig. 6). By performing the same analysis as depicted in Fig. 7, the architecture of loops 1, 2, and 7 in EndoBT were found to be most similar to those observed in EndoT, EndoH, and EndoF₁, strongly suggesting that the enzyme is an endoglycosidase specific for high-mannose-type oligosaccharides. We therefore determined the ability of EndoBT to hydrolyze biantennary complex-type N-linked glycans and/or high-mannose-type N-linked glycans from IgG1 antibodies. As predicted from our structural analysis, these assays revealed that EndoBT

hydrolyzes high-mannose-type IgG1 but not biantennary complex-type IgG1 (Fig. 9e).

Altogether, our data provide critical insights into the structural determinants of complex-type N-linked glycan specificity of EndoS, a key event for *S. pyogenes* to evade the host immune system. Moreover, the identification of the loops surrounding the carbohydrate binding site that are responsible for the bianntenary complex-type N-linked glycan specificity of EndoS, together with the structural comparison of these loops in the framework of GH18 endoglycosidases with different glycan specificities,

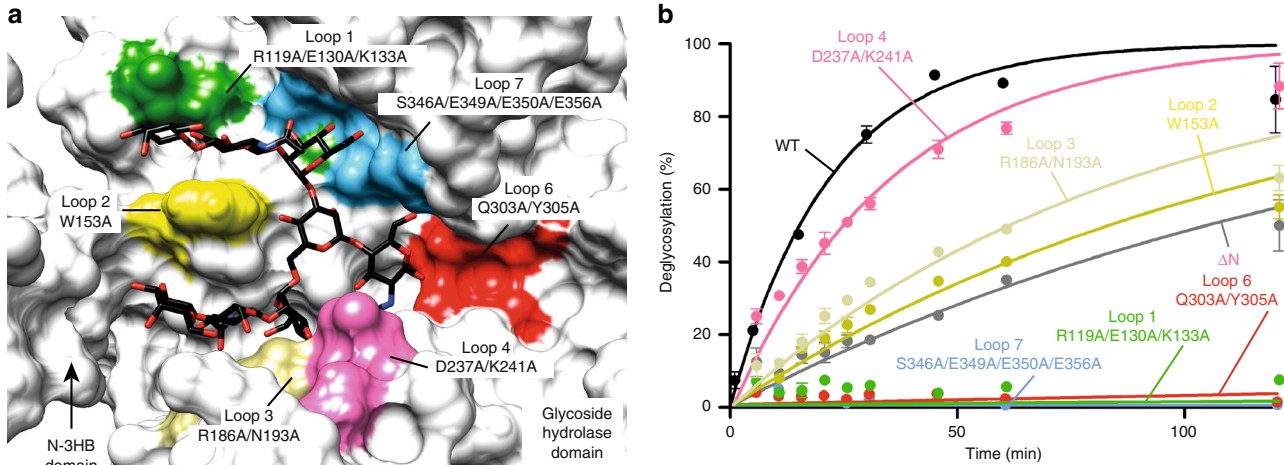

**Fig. 6** Structural basis of EndoS endoglycosidases specificity. **a** Surface representation of EndoS$_{D233A/E235L}$ in the complex, with G2 product showing the alanine mutations performed in loop 1 (green), loop 2 (yellow), loop 3 (gold), loop 4 (pink), loop 6 (red), and loop 7 (light blue) in the glycosidase domain (gray). **b** Hydrolytic activity of EndoS wild-type and mutants against Rituximab is shown, as determined by LC-MS analysis. The color code is equivalent to that displayed in **a**. ΔN-3HB-EndoS is in gray. Data points reflect the mean of two separate measurements, error bars indicate standard deviation

| | Loop-1 120–145 | Loop2 151–158 | Loop3 185–206 | Loop4 235–247 | Loop5 281–289 | Loop6 304–306 | Loop7 347–380 |
|---|---|---|---|---|---|---|---|
| EndoS | | | | | | | |
| EndoF₃ | | | | | | | |
| EndoF₁ | | | | | | | |
| EndoH | | | | | | | |
| EndoT | | | | | | | |
| EndoBT | | | | | | | |

**Fig. 7** Structural basis of GH18 endoglycosidases specificity. Structural comparison of the loops surrounding the active site of GH18 family enzymes with endo-N-acetyl-β-D-glucosaminidase activity. Oligosaccharide moieties that interact with each loop in the crystal structure of EndoS$_{D233A/E235L}$-G2 complex and EndoF$_3$ are marked with red squares

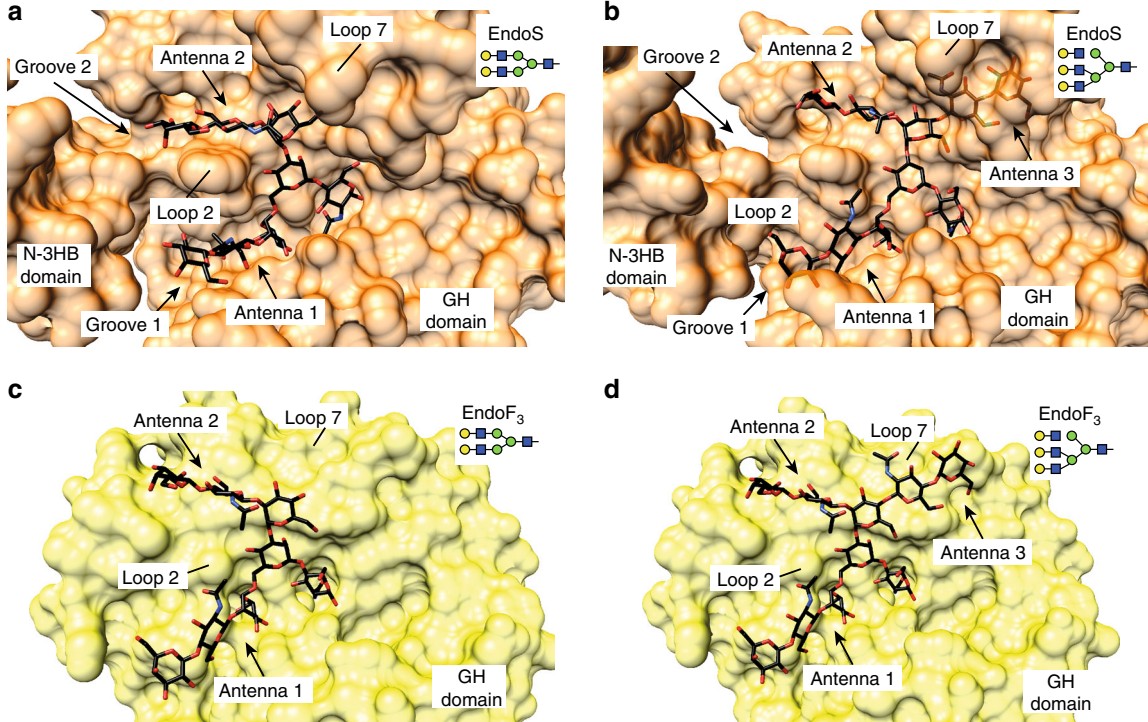

**Fig. 8** Structural basis of EndoS and EndoF$_3$ specificity. **a** Crystal structure of the EndoS$_{D233A/E235L}$-G2 complex (6E3N). **b** Superimposition of EndoS$_{D233A/E235L}$-G2 and the triantennary complex-type oligosaccharide product. **c** Crystal structure of the EndoF$_3$-G2 product complex (1EOM). **d** Triantennary complex-type oligosaccharide product docked into EndoF$_3$

provides the basis for the bioengineering of endoglycosidases towards more efficient and customizable chemoenzymatic synthesis of therapeutic monoclonal antibodies.

## Methods

**Purification of EndoS wild-type and EndoS mutants.** EndoS$_{D233A/E235L}$-CPD, EndoS wild-type and EndoS mutants were purified as previously described with the following modifications[18]. Single-point mutations were made using the FastCloning[36] method, and full sequences were confirmed by GeneWiz (https://www.genewiz.com). BL21(DE3) (Novagen) cells transformed with the corresponding modified form of the pCPD vector (pCPD-L) containing the C-terminal fusion protein from *Vibrio cholerae* MARTX toxin cysteine protease domain (CPD)[37] were grown in Luria broth (LB) medium supplemented with 50 μg ml$^{-1}$ ampicillin. Cultures were grown at 37 °C to an OD$_{600}$ of 0.6–0.8, at which point the temperature was lowered to 22 °C over 1 h. Induction was triggered with 0.5 mM isopropyl-D-1-thio-galactopyranoside (IPTG) at 22 °C overnight. Cells were harvested by centrifugation and lysed by sonication using 50 mM Tris-HCl pH 7.5, 500 mM NaCl, 10% glycerol (solution A) containing protease inhibitors (Complete EDTA-free, Roche). The supernatant was applied to a HisTrap Chelating column (1 ml, GE HealthCare) equilibrated with solution A. The column was then washed with solution A until no absorbance at 280 nm was detected. For EndoS$_{D233A/E235L}$, elution was performed with a linear gradient of 40–500 mM imidazole in 50 mM Tris-HCl pH 7.5, 500 mM NaCl at 1 ml min$^{-1}$. The C-terminal CPD tag of EndoS$_{D233A/E235L}$ was not hydrolysed and this enzyme was further purified by size-exclusion chromatography using a Superdex 200 10/300 GL column (GE Healthcare) equilibrated in 20 mM Tris-HCl pH 7.5, 50 mM NaCl. The eluted protein was concentrated to 10 mg ml$^{-1}$ using Amicon-15 centrifugal filter (Millipore) unit, with a molecular cut off of 100 KDa at 4000×$g$. The resulting preparation displayed a single protein band when run in 10% SDS/PAGE stained with Coomassie Blue. EndoS wild-type and the other EndoS mutants were treated with 1 mM phytic acid overnight on the HisTrap column to hydrolyze the CPD tag. Proteins were then buffer exchanged into PBS, pH 7.4 and further purified by size-exclusion chromatography in a Superdex 200 10/300 GL column (GE Healthcare) equilibrated in PBS, pH 7.4. The eluted proteins were concentrated to at least 0.2 mg ml$^{-1}$ using the same procedure explained above, diluted to 50 nM stocks, and then stored at 4 °C.

**EndoS$_{D233AE235L}$-G2 crystallization and data collection.** The crystal of EndoS$_{D233A/E235L}$-G2 was obtained by mixing 0.25 μl of the protein (10 mg ml$^{-1}$) in 20 mM Tris-HCl pH 7.5, 50 mM NaCl and 2.5 mM G2 product with 0.25 μl of a mother liquor containing 100 mM sodium HEPES/MOPS pH 7.5, 100 mM amino

acid mixture (L-Na-glutamate, alanine (racemic), glycine, lysine HCL (racemic), serine (racemic)), 20% (w/v) PEG 500 MME and 10% (w/v) PEG 20,000 using the sitting drop vapor diffusion method. The crystal appeared after 21 days and was washed with the mother liquor and frozen under liquid nitrogen. X-ray diffraction data was collected on a EIGER X 9M photon-counting area detector (2000 Hz max. frame rate) at the microfocus PROXIMA 2–A beamline (λ = 0.9801 Å—SOLEIL, France, see Supplementary Table 1 for details). Data were integrated and scaled with XDS following standard procedures[38].

**EndoS$_{D233AE235L}$-G2 structure determination and refinement.** Structure determination of EndoS$_{D233AE235L}$-G2 was resolved using as a template the previously reported EndoS structure (unmodified PDB 4NUZ)[18] and molecular replacement methods implemented in Phaser[39] and the PHENIX suite[40]. Model rebuilding was carried out with Buccaneer[41] and the *CCP4* suite[42]. The final manual building was performed with Coot[43] and refinement with phenix.refine[44]. The structure was validated by MolProbity[45]. Data collection and refinement statistics are presented in Supplementary Table 1. Atomic coordinates and structure factors have been deposited with the Protein Data Bank, accession code 6EN3. Molecular graphics and structural analyses were performed with the UCSF Chimera package[46].

**SAXS measurements.** Synchrotron X-ray scattering data for recombinant purified EndoS$_{D233A/E235L}$ were collected on the B21 beamline of the Diamond Light Source, UK. Data collection was performed in batch mode. The sample volume loaded was 30 μL (1.5 mm diameter capillary with 10 μm wall thickness). Data were collected using a Pilatus2M detector (Dectris, CH) at a sample-detector distance of 3914 mm and a wavelength of λ = 1 Å. The range of momentum transfer of $0.1 < s < 5$ nm$^{-1}$ was covered ($s = 4\pi\sin\theta/\lambda$, where $\theta$ is the scattering angle). Scattering patterns were measured with a 0.5-s exposure time (18 frames) for protein samples at a minimum of three different protein concentrations ranging from 0.5 to 4 mg ml$^{-1}$. To check for radiation damage, 20–50 ms exposures were compared; no radiation damage was observed. EndoS at 1, 2, and 4 ml min$^{-1}$ in 50 mM Tris, pH 7.5, 100 mM NaCl and 2% glycerol were incubated with G2 product at 0.625, 1.25, and 2.5 mM before data collection for 30 min. Data were processed and merged using standard procedures by the program package ScÅter[47] and PRIMUS[48]. Concentration dependent effects were not observed by comparing the curves obtained from the three different concentrations. Scattering curves at multiple concentrations were then scaled and merged into a single scattering curve for further analysis. Using CRYSOL[49], we fitted the SAXS data in the presence and absence of the oligosaccharide G2 product to the solution-scattering profile calculated from the crystal structure (6EN3). Analyses of potential conformational transitions were conducted using an elastic network procedure implemented in the program SREFLEX[50]. The better fit obtained using SREFLEX and the normalized Kratky plot suggest the protein show

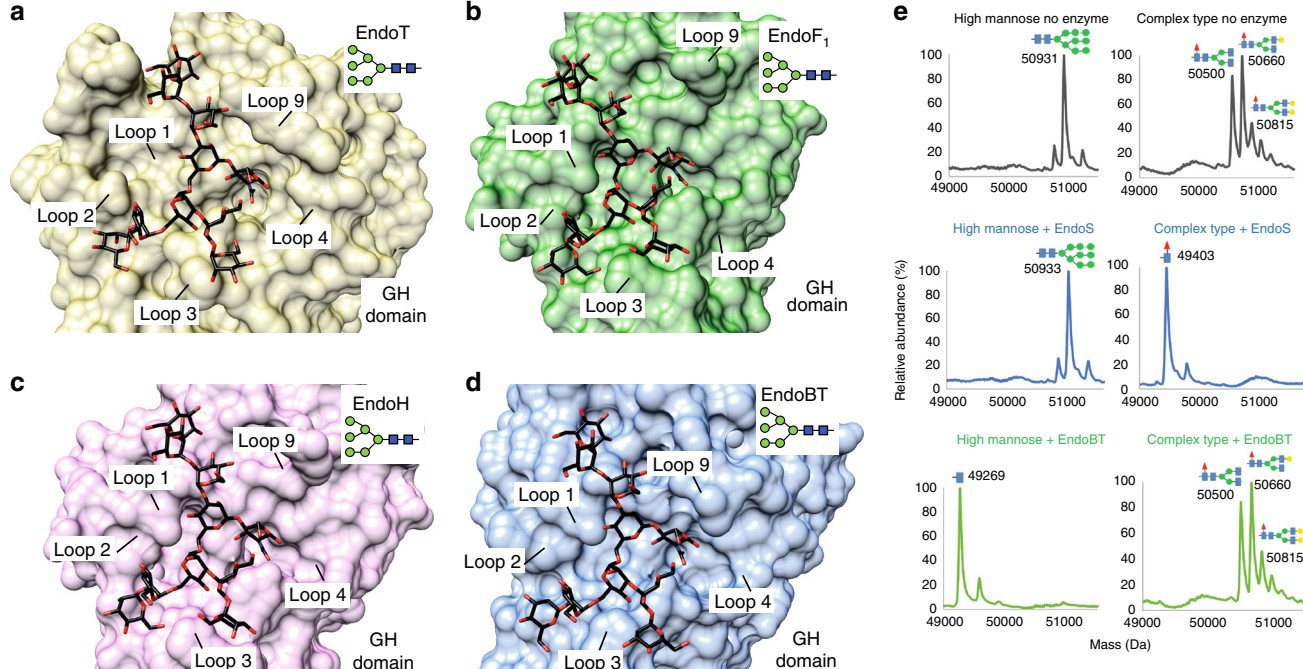

**Fig. 9** Structural basis of EndoT, EndoH, EndoF₁ and EndoBT specificity. Molecular models of the docked GlcNAcMan₉ oligosaccharide in the binding site of **a** EndoT **b** EndoF₁ **c** EndoH and **d** EndoBT. **e** LC-MS analysis of the glycosyl hydrolase activity of EndoS (middle panel) and EndoBT (bottom panel) on high-mannose-type and complex-type IgG1. A negative control of the reaction is shown in the upper panel

some flexibility in solution[51]. The maximum dimensions ($D_{max}$), the interatomic distance distribution functions ($P(r)$), and the radii of gyration ($R_g$) were computed using GNOM[52]. The molecular mass was determined using ScÅter[47,53]. The low-resolution structures of EndoS$_{D233A/E235L}$ in presence and absence of G2 product were calculated ab initio by using GASBOR[54]. The results and statistics are summarized in Supplementary Table 2.

**Chemical synthesis of G2 product**. The desialylated complex-type glycan (CT, **5**) was obtained from sialylglycoprotein (SGP, **1**) isolated from egg yolk (Supplementary Fig. 7)[55]. SGP (100 mg, 34.9 μmol) was dissolved in phosphate buffer (50 mM, pH 6.5, 5 ml) and then treated with wild-type EndoM endoglycosidase (500 μg, 0.1 μg μl⁻¹) at 37 °C overnight. Upon monitoring using high pH anion exchange chromatography (HPAEC), the reaction was deemed complete and the crude purified on reverse-phase HPLC. The glycan-positive fractions were desalted on a Sephadex G10 gel filtration column with DI H₂O as the eluent. The pooled glycans comprised a mixture of sialylated (**2**), monosialylated (**3**) and mono-sialylated degalactosylated (**4**) glycoforms which were further purified using anion exchange chromatography to give the pure sialylated glycoform (SCT, **2**). The sialylated glycoform (**2**, 20 mg, 9.9 μmol) was dissolved in citrate buffer (50 mM, 5 mM CaCl₂, 200 μl) and incubated at 37 °C in the presence of sialidase (50 U). The reaction was monitored using HPAEC and was complete in 6 h followed by treatment with Dowex resin (H⁺ form). The crude was centrifuged and the supernatant desalted using a Sephadex G10 gel filtration column eluting with DI H₂O. The glycan fractions were pooled then lyophilized to furnish the product asialo complex-type glycan (**5**) as a white powder (13 mg, 92%). The product was characterized using HPAEC and ESI mass spectrometry. ESI MS: calcd. $M =$ 1437.51; found (m/z): 1438.52 $[M + H]^+$, 719.76 $[M + 2H]^{2+}$.

**Thermal stability assays**. Melting temperatures for purified proteins were determined using differential scanning fluorimetry[56]. EndoS$_{WT}$ and EndoS$_{98-995}$ were diluted to a final concentration of 0.5 mg mL⁻¹ in PBS pH 7.4, and mixed with 5000× Sypro Orange (Sigma) to a final concentration of 5× in a 96 White TempPlate with semi-skirt (USA Scientific). Melting curves were measured on an iQ5 Multicolor Real Time PCR Detection System (Bio-Rad). Data were obtained from 25 to 95 °C with 1 °C intervals and 1-min dwell time at each temperature before measuring fluorescence.

**EndoS and EndoBT enzymatic activity assay**. Reactions were set up using 5 nM EndoS or EndoS mutants, or 100 nM EndoBT and 5 μM Rituximab or high-mannose-type IgG1 in PBS pH 7.4 at room temperature. Rituximab, a chimeric anti-human CD20 monoclonal antibody approved for treatment of B-cell lymphoma in adults, is produced in mammalian cell (Chinese Hamster Ovary) culture with the most abundant glycoforms being G0F, G1F, and G2F (antibody purchased

from Premium Health Services, Inc.)[57]. At various time intervals, 10 μl aliquots of the reaction were taken in duplicate and quenched with 1.1 μl of 1% trifluoroacetic acid. The quenched reaction was then mixed with 50 mM TCEP, and analyzed by LC-MS using an Accela LC System attached to a LXQ linear ion trap mass spectrometer (Thermo Scientific, Waltham, MA). Relative amount of the substrate and the hydrolysis products were quantified after deconvolution of the raw data and identification of the corresponding MS peaks using BioWorks (Thermo Scientific, Waltham, MA). The data were plotted in GraphPad Prism, and fit with a one-phase exponential decay curve.

**Structural analysis and sequence alignment**. Structure based sequence alignment analysis were performed using Chimera[46]. Protein pocket volume was calculated using HOLLOW[58]. $Z$-score values were produced by using DALI[29]. Domain interface analysis was performed using PISA[19]. Conserved and similar residues were labeled using BoxShade server (http://embnet.vital-it.ch/software/BOX_form.html).

**Molecular docking calculations**. The first GlcNAc (−1) and the last Neu5Ac (6 and 10) residues of the S2G2 substrate; and the Man₉GlcNAc product were modeled using GLYCAM-Web website (Complex Carbohydrate Research Center, University of Georgia, Athens, GA (http://www.glycam.com))[59]. Ligand docking was performed using AutoDock Vina employing standard parameters[60].

**Purification of EndoBT**. EndoBT (BT_3987 (*B. thetaiotaomicron* VPI-5482)) in pSpeedET vector was purchased from DNASU plasmid repository (https://dnasu.org/DNASU/Home.do). EndoBT was expressed in BL21(DE3) (Novagen) in LB medium supplemented with 50 μl ml⁻¹ ampicillin. Cultures were grown at 37 °C to an OD$_{600}$ of 0.6–0.8, at which point the temperature was lowered to 22 °C during 1 h. Induction was triggered with 0.5 mM IPTG at 22 °C overnight. Cells were harvested by centrifugation and lysed by sonication using PBS, pH 7.4 and 10% glycerol (solution A), containing protease inhibitors (Complete EDTA-free, Roche). The supernatant was applied to a HisTrap Chelating column (1 ml, GE HealthCare) equilibrated with solution A. The column was then washed with solution A until no absorbance at 280 nm was detected. Elution was performed with a linear gradient of 40–500 mM imidazole in PBS at 1 ml min⁻¹. EndoBT was further purified by size-exclusion chromatography using a Superdex 200 10/300 GL column (GE Healthcare) equilibrated in PBS, pH 7.4. The eluted protein was stored at −80 °C.

**Purification of high-mannose IgG1**. CD4-induced IgG1 plasmid[61] was transiently expressed in HEK293T cells (ATCC) using polyethylenimine as transfection reagent, and in the presence of kifunensine. Kifunensine is a potent inhibitor of the mannosidase I enzyme, which drastically reduces the complexity of the

carbohydrates by blocking the oligosaccharide at the stage of high-mannose type[62]. After transfection, cells were cultured for 96 h in Free-style F17 medium supplemented with GlutaMAX and Geneticin (Thermo FisherScientific). High-mannose IgG1 was purified from culture supernatants by protein A chromatography using 20 mM sodium phosphate buffer pH 7.0 as binding buffer and 100 mM sodium citrate buffer pH 3.0 as elution buffer. All the fractions were neutralized with 1 M Tris pH 9.0, pooled and dialyzed against PBS at pH 7.5.

**Data availability**. Atomic coordinates and structure factors data that support the findings of this study have been deposited in the PDB with the accession code 6EN3 (https://www.rcsb.org/structure/6EN3). All other data that support the findings of this study are available from the corresponding authors on reasonable request.

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

## Acknowledgements

This work was supported by the MINECO Contract BFU2016-77427-C2-2-R and Severo Ochoa Excellence Accreditation (SEV-2016-0644) (to M.E.G.), Juan de la Cierva Program IJCI-2014-19206 (B.T.). We acknowledge Diamond Light Source (proposals mx15304), SOLEIL (proposal 20150773), and iNEXT (proposals 1618/2538) for providing access to synchrotron radiation facilities. We gratefully acknowledge all members of the Structural Glycobiology Group (CIC bioGUNE, Spain) for valuable scientific discussions.

## Author contributions

E.J.S., L.-X.W. and M.E.G., conceived the project. B.T., E.K., J.O., and A.M., performed the experiments. B.T., E.K., J.O., A.M., E.J.S., L-X.W., and M.E.G., analyzed the results. B. T., E.K., E.J.S., and M.E.G., wrote the manuscript.

## Additional information

**Competing interests:** The authors declare no competing interests.

