## [Peer Review File · Nature Communications]

Reviewers' comments:

Reviewer #1 (Remarks to the Author):

The manuscript by Trastoy et al. describes primarily the crystal structure of a complex of the IgG-specific endoglycosylase S (EndoS) with the G2 product of its reaction with the glycosylation on Asn127 of the Fc domain of IgG. The interactions between enzyme and product are described in detail. SAXS studies are used to show that there are no large-scale conformational changes in solution upon product binding. Mutations in loops shown from the crystal structure to be important for enzyme-product interactions are made and their effect on enzymatic activity against the model IgG Rituximab are analysed, highlighting the importance of loops 1, 6 and 7.

In summary, while the results are undeniably interesting, it is uncertain to me to what extent they would influence thinking in the field. The results are similar to those obtained for EndoF3 more than 15 years ago. I feel that detailed comparison with EndoF3, which has very much overlapping substrate specificity, is lacking. How do the differences between EndoS and EndoF3 explain how EndoF3 accommodates the same product as EndoS while also accepting triantennary glycosylations? Finally, the manuscript is in places poorly written, with poor figure legends and inconsistent reference to figures. I would recommend either that the work is sent to a more specialised journal or resubmitted here with major revision.

Some comments to help improve the manuscript:

I think that the picture of the glycosylation in Fig. S2 should be in the main manuscript.

line 49: "between the first two N-acetylglucosamine residues"

line 53: "glycoforms and"

line 74: how was the volume of the G2 glycan binding site estimated?

lines 84 and onwards: the numbering of the saccharide rings used in this description is not shown in any figure.

line 96: "protrude away"

line 99: "Figure S5"

line 101: what is the S2G2 substrate?

line 108: "comprised" is unnecessary

lines 108-110: It's unnecessary to write "Ala" at the end of every mutation, as it was specified in the previous sentence that they are all Ala mutations.

line 121: I would include references to Figs. 2d and 3a here. Why do references to the original unbound crystal structure (ref. 18) appear for the first time here?

line 121: Supplemental Figure S6 is used to support a statement about the position of Trp153 at the bisection point of the two binding grooves. However the main aim of this figure seems to be to describe differences between the bound and unbound structures. No mention of this is made in the main text. Furthermore, Figure S6 is in contradiction to Figure S4, where the crystal structure of the complex is superposed with the envelope of both the bound and unbound forms to assert that there is no difference in conformation. In Fig. S6b one can clearly see a change in the conformation of two domains. The fitting of both forms to the SAXS data of both forms (i.e. 4 comparisons) should be

investigated. What is the chi-squared value for the crystal structures(s) using CRY SOL?
line 124: what program produced the Z-score? Was it DALI? If so, it should be mentioned and cited.

Fig. 2: The choice of bright, reflective colours for the molecular surface makes it very hard to see the bound product. It's not clear what the point of panel b) is.

Online methods:

line 262: the reference 21 should be moved to just after "FastCloning method" on line 261 and an address should be given for GeneWiz.

line 265: I guess 50 micrograms/ml is meant

line 275: Coomassie Blue

line 279: What happened after the protein was loaded onto the Superdex column? I guess it was eluted in some way? How was it concentrated and stored?

line 283: sodium

line 284: amino acid mixture (L-Na-glutamate,... lysine HCl)

line 290: How were the data scaled, with XSCALE or AIMLESS in CCP4?

line 294: Was 4NUZ used unmodified as a search model, or was it split into domains?

line 301: SAXS is not diffraction.

line 303: batch mode

line 305: The description of SAXS data collection is unsatisfactory. What was the sample size? Capillary volume? Temperature? Was the sample flowed or oscillated to avoid radiation damage? Was the $I(0)$ normalised?

line 311: Were any attempts made to do ab initio reconstruction using DAMMIN/DAMMIF?

line 319: The lack of conformational change is a result and should not be mentioned only in the Online Methods section, but in the main text.

The supplementary/supplemental/supporting material (the authors use all three terms) is a bit sloppily prepared, with above all inadequate figure legends.

Table S1: The table seems to have been copied from phenix.table_one without too much afterthought.

The space group is more simply written $P2(1)$, with the 1 as a subscript. The use of two decimal places for unit cell dimensions and for the B-factors is not necessary. The difference between Rmodel and Rfree seems rather small for a 2.9 Å structure: were the Rfree flags used to cross-validate the unbound structure transferred to the dataset for the bound structure presented here? It isn't stated anywhere how many reflections were used for the Rfree calculation.

Table S2: The symbol \leq should presumably be \pm . There seems to be an extra digit, "1" after the end of the second Porod volume estimate. The units of $I(0)$ cannot be reciprocal Ångström, as this is the unit of q , on the x-axis of the scattering curve. $I(0)$ is on the y axis and has arbitrary units unless normalised using a known standard (was this done?) Likewise, why is the unit reciprocal cm used for the $I(0)$ estimate from Guinier? "Comparison" should be "comparison". Why is the CRY SOL chi-squared value so high? Did the authors consider EOM? The program "SCATER" should be spelled "SCÅTTER"

In Fig. S1, the figure gives the misleading impression that the glycosylations are highly exposed on the exterior of the Fc domain, when they are in fact on the inside. While I appreciate that this can be hard to draw, it should at least be indicated in the figure legend. It should be explained what the right-hand reactive group of the molecule used to attach uniform glycosylations to IgG is. The cleavage point leading to the G2 product should be indicated. The symbols for the different types of saccharide unit should be defined. There should be a legend for this figure!

Fig. S2 legend: "996-1191 residues" should read "residues 996-1191". The sequence should be numbered.

In Fig. S4, the same colour scheme should be used for the ab initio reconstructions as for the scattering curves and $P(r)$ functions. As it is, the orange colour is used for the complexed protein in panel a and for the uncomplexed protein in panel b.

In Fig. S6, the legend states that the complexed structure is orange in both panels a and b, when it is only orange in panel A. What colour scheme is used for panel b? Is the uncomplexed structure still light grey?

In Fig. S7, the structure-based sequence alignment cannot be completely correct, because e.g. the second of the catalytic residues marked by red dots is not conserved in EndoF1 and EndoBT.

Fig. S8 seems to show a biochemical synthesis rather than a chemical one.

Reviewer #2 (Remarks to the Author):

This manuscript describes the structure of a complex between the IgG-specific endoglycosidase, EndoS, and a biantennary complex type N-glycan. The resulting structure describes the manner by which the endoglycosidase recognizes the branched glycan product in the enzyme active site.

While the manuscript is an effective structure presentation of the enzyme-glycan product complex, there are some deficiencies that need to be resolved in the description of the complex, including:

- 1) The authors completely under-represent the fact that they already published the overall structure of apo-EndoS (reference 18, Trastoy et al (2014) PNAS). The reference of this prior paper was cryptically cited during the structure description. However, it is so buried that a cursory read of the present paper would have left a reader with an impression that this was the first structural description of this protein. As such, the novelty of the present paper is the structure of the glycan substrate complex, not the initial description of the protein. The details of the complex are well described in the manuscript and the mutagenesis data, coupled with kinetic analysis, are an effective description of how the recognition of the biantennary complex glycan structure is achieved.
- 2) In Fig 3C the authors attempt to make a structural comparison with other endoglycosidase structures, but this visual representation is not effective. The nature of the isolated loop regions in cartoon representation in the illustration provide no meaningful insight to the differences in

structural basis of substrate specificity. The authors state that “Residues involved in the interaction with the conserved Man β 1–4GlcNAc β 1–4GlcNAc β 1 13; core are essentially preserved ...” This conservation in interactions with these glycan residues are not evident from this figure. What are the PDB files used in this comparison along with the respective citations of the structures? How many of the structures have substrate complexes? The authors make cursory statements about the comparison with the substrate complex for EndoF3. The previous PNAS paper presented a structural comparison of EndoS with the EndoF3-glycan complex, but the present manuscript needs to show a detailed structural comparison of the EndoF3 complex with the EndoS-glycan complex. What is novel about the present glycan structural complex by comparison to the prior work on EndoF3?

3) In the prior PNAS paper the authors present a model of the docked EndoS complex with IgG and how the EndoS accessory domains enable the insertion of the glycan into the active site. Is this model still relevant based on the present glycan complex?

4) The authors model in the GlcNAc(-1) and NeuAc(+6) and (+10) monosaccharides in Fig. S5, but never describe how that modeling was performed nor is it described in either the legend to Fig. S5 or the body of the paper that these monosaccharides were modeled and not a part of the empirically derived structure.

5) It would be very helpful to show the positions of the catalytic residues that were mutated and provide a model for the catalytic mechanism that resulted in the formation of the enzymatic product complex that is present in the active site. This would best be presented in conjunction with Fig. 2D.

6) In regard to the N-3HB domain, the authors state that: “these data suggest that the N-3HB domain is a key element involved not only in N-linked glycan substrate recognition but also in EndoS-Fc fragment interaction.” The data do not really suggest these conclusions. An equally likely interpretation is that the N-3HB domain stabilizes the GH domain structure. The IgG domain would be expected to engage the active site from the opposite side of the GH domain from the N-3HB domain and would not likely directly interact with the IgG domain.

Other minor issues:

1) Please move the Man+7 label out into white space in Fig 2D.

2) Show residue D233A in Fig 2D.

Overall, the manuscript is a nice structural presentation of the enzyme-glycan product complex for EndoS and provides tremendous insights into how this enzyme is able to recognize biantennary complex glycan structures. Additional detail is required to provide insights into the novelty of this structure in compared to the prior PNAS paper and to the structure of the EndoF3-glycan complex. No additional experimentation is required, but a more effective presentation of comparative data with other structures would greatly improve the manuscript.

Reviewer #3 (Remarks to the Author):

IgG glycans regulate antibody effector functions. EndoS from *Streptococcus pyogenes* cleaves specifically complex-type N-linked glycans of IgG and contributes to the immune evasion of the bacterium. Interestingly, EndoS glycosynthase variants can be used to transfer pre-defined complex type N-glycans to intact IgG, which could be very useful to generate therapeutic antibodies with the desired glycoforms. With the aim to better understand the substrate specificity of this enzyme, Trastoy and collaborators have crystallized an inactive variant of endoS in complex with its oligosaccharide product (G2 glycoform). Following the identification of EndoS domains interacting with the G2 glycoform, authors have performed key mutations of amino acid residues to demonstrate their involvement in the recognition of the glycan product. Finally, by comparing the 3D structure of EndoS domains with those of other glycosylhydrolases members of the GH18 family, authors proposed that the substrate specificity of EndoS relies on the unique groove 2 of the enzyme. Although the manuscript is clearly written, of great interest and provides several novel findings, additional experiments are required to fully support the conclusions made by authors:

Major comments:

1) My main concern is related to the last part of the manuscript. Based on the functional data depicted in figure 3 and comparison of endoglycosidase 3D structures, authors proposed that the EndoS specificity for complex-type N-linked oligosaccharide relies on its exclusive groove 2 structure (loops 1, 2 and 7), which is significantly different than those of other GH18 endoglycosidases. But authors did not directly demonstrate it. Authors mentioned (Lines 141-143 and Fig. 3C) that loops 1 and 2 of high-mannose-type specific GH18 endoglycosidases are “ordered” and adopt “a beta-hairpin conformation” but not that of endoS. However, a counterexample of this is the lack of order of loops 1 and 2 of EndoT, which is also a high-mannose-type specific endoglycosidase. Likewise (Line 144), authors mentioned that the Loop7 is “markedly shorter” in high-mannose-type specific GH18 endoglycosidases as compared to that of EndoS. However, EndoF3 that recognizes complex-type N-glycans, also have a very short loop 7. A direct and more convincing proof of the involvement of loop 1, 2 and 7 roles in the substrate specificity of EndoS would be the substitution or introduction of residues (or even entire loops) from corresponding loops in high-mannose-type-specific GH18 endoglycosidases (Ex. EndoH), following by the measurement of binding of different glycan products (G2, High-mannose-type glycans,...) or, if possible, measurement of activity of EndoS variants on different substrates.

Minor comments

- 1) Figures 3a and b- mutations are presented in terms of amino acids (ex. R119A/E130A/K133A) whereas the corresponding text (Lines 104-128) refers to Loops, making the demonstration hard to follow. Please clearly present the Loops also in both figures.
- 2) Figure 3c: For Endo S, Loop-1 is indicated to interact with two of the core mannoses whereas Figure 3a suggest that Loop-1 interacts with terminal glucosamine and galactose. Please, correct or explain this discrepancy.
- 3) Figure 3c: Loops 3, 4, 5 and 6 are not at all commented into the text, compromising the interest of presenting them into the table. I would suggest moving them in supplementary data
- 4) Figure 3b- A presentation of mutant ΔN is lacking in the legend.

- 5) Line 53- typo « glycoformsand and »
- 6) Line 69- define or explain « hybrid Ig domain »
- 7) Line 95- Correct: “residues of each arm adopt two “alternative” (instead of different) conformations” and precise “into the crystal”. Likewise, in Figure S5- Correct “the two alternative conformations of G2 product”.
- 8) Line 125- the mode of interaction of the SpA C domain (similar to endoF N-3HB) with IgG or IgM is not clearly explained. The reader cannot even discriminate between protein-protein or protein-glycan interfaces ...
- 9) Figure S1- For a better understanding, please add some explanations on the fig S1 legend. I also don't understand why the item “Immune evasion” is below the second arrow and not below the first arrow.
- 10) Figure S6- Line 110- Correct “a” and “b” instead of “A” and “B”.
- 11) Figure S6A- Loop 2 should be highlighted in the figure.
- 12) Lines 138-139- steric hindrance is hypothesized for EndoS, and compared to tridimensional permissiveness in EndoF3 or EndoH, but no 3D illustration of EndoF3 or H groove 2 corroborate the comments (as Figures 3c focuses on EndoS and Figure S7 presents truncated loops in a series of Endo enzymes). This should be illustrated.

Reviewer #1:

1. The manuscript by Trastoy et al. describes primarily the crystal structure of a complex of the IgG-specific endoglycosylase S (EndoS) with the G2 product of its reaction with the glycosylation on Asn127 of the Fc domain of IgG. The interactions between enzyme and product are described in detail. SAXS studies are used to show that there are no large-scale conformational changes in solution upon product binding. Mutations in loops shown from the crystal structure to be important for enzyme-product interactions are made and their effect on enzymatic activity against the model IgG Rituximab are analysed, highlighting the importance of loops 1, 6 and 7. In summary, while the results are undeniably interesting, it is uncertain to me to what extent they would influence thinking in the field. The results are similar to those obtained for EndoF3 more than 15 years ago. I feel that detailed comparison with EndoF3, which has very much overlapping substrate specificity, is lacking. How do the differences between EndoS and EndoF3 explain how EndoF3 accommodates the same product as EndoS while also accepting triantennary glycosylations?

ANSWER: We very much appreciate that you consider our study of undeniable interest, and all the constructive comments/suggestions you made to improve the article.

In the new version of the manuscript, we have now included an extensive and detailed comparison and discussion on the structural differences of the EndoS and EndoF₃ glycosidase domain, and the importance of these differences in defining the glycan specificity of each enzyme. Please see ‘Structural comparison of EndoS with GH18 family of endoglycosidases’ in the new ‘Results and Discussion’ section, and new Figures 8 and 9.

2. Finally, the manuscript is in places poorly written, with poor figure legends and inconsistent reference to figures. I would recommend either that the work is sent to a more specialised journal or resubmitted here with major revision.

ANSWER: The original manuscript was transferred from another NPG journal with a considerably more condensed format. In the process of reshaping the manuscript, the comments made by the reviewers have been invaluable.

3. Some comments to help improve the manuscript:

I think that the picture of the glycosylation in Fig. S2 should be in the main manuscript.

ANSWER: We agree with the reviewer. We have included the picture of the glycosylation pathway as Figure 1 in the main manuscript.

4. line 49: "between the first two N-acetylglucosamine residues"

ANSWER: We have modified the text accordingly.

5. line 53: "glycoforms and"

ANSWER: We have modified the text accordingly.

6. line 74: how was the volume of the G2 glycan binding site estimated?

ANSWER: We have used the program HOLLOW to calculate the volume of the G2 glycan binding site. We have included the reference in the main text.

Ho, B.K., Gruswitz, F. HOLLOW: generating accurate representations of channel and interior surfaces in molecular structures. BMC Struct Biol 2008, 8:49 (2008).

7. lines 84 and onwards: the numbering of the saccharide rings used in this description is not shown in any figure.

ANSWER: We agree with the reviewer. The numbering of the saccharide rings has been modified and now included in Figure 3d, Figure 4b,d, Figure 5a,b, and Supplementary Figure 3c,d.

8. line 96: "protrude away"

ANSWER: We have modified the text accordingly.

9. line 99: "Figure S5"

ANSWER: We have modified the text as 'Supplementary Fig. 5'.

10. line 101: what is the S2G2 substrate?

ANSWER: We agree with the reviewer. A representation of S2G2 substrate has been included in the new Figure 5a.

11. line 108: "comprised" is unnecessary

ANSWER: We have modified the text accordingly.

12. lines 108-110: It's unnecessary to write "Ala" at the end of every mutation, as it was specified in the previous sentence that they are all Ala mutations.

ANSWER: We agree with the reviewer. We have modified the text accordingly.

13. line 121: I would include references to Figs. 2d and 3a here.

ANSWER: We agree with the reviewer. In the new version of the manuscript, we have modified the text accordingly.

14. Why do references to the original unbound crystal structure (ref. 18) appear for the first time here?

ANSWER: Please see answer to point 2. In that sense, (i) we have included the original unliganded crystal structure from reference 18 in the new ‘Introduction’ and ‘Results and Discussion’ sections of the main text and (ii) extended the comparison between both crystal structures in the new ‘Results and Discussion’ section and new Supplementary Figures 3 and 4.

15. line 121: Supplemental Figure S6 is used to support a statement about the position of Trp153 at the bisection point of the two binding grooves. However, the main aim of this figure seems to be to describe differences between the bound and unbound structures. No mention of this is made in the main text.

ANSWER: We agree with the reviewer. In the new ‘Results and Discussion’ section we carefully describe the differences between EndoS-G2 complex and the unliganded form of Δ N-3HB-EndoS structures. Please see also the new Supplementary Figure 3.

16. Furthermore, Figure S6 is in contradiction to Figure S4, where the crystal structure of the complex is superposed with the envelope of both the bound and unbound forms to assert that there is no difference in conformation. In Fig. S6b one can clearly see a change in the conformation of two domains.

ANSWER: We observed a similar fit of the SAXS data in the presence and absence of the G2 product to the solution scattering profile calculated from the crystal structure of EndoS_{D233A/E235L}-G2 product complex, suggesting that the overall shape of the enzyme remains unchanged. However, conformational flexibility of EndoS is suggested by (i) the normalized Kratky plot of EndoS_{D233A/E235L} (please see new Supplementary Figure 4b), and (ii) the structural comparison of the X-ray crystal structures of EndoS_{D233A/E235L}-G2 complex and the Δ N-3HB-EndoS (please see new Supplementary Figure 4c).

17. The fitting of both forms to the SAXS data of both forms (i.e. 4 comparisons) should be investigated.

ANSWER: We could only fit the solution scattering profile calculated from the X-ray crystal structure of EndoS_{D233A/E235L}-G2 and the experimental scattering curves obtained from EndoS_{D233A/E235L} and EndoS_{D233A/E235L}-G2 in solution. There is no X-ray crystal structure of the unliganded form of full length EndoS_{D233A/E235L}.

18. What is the chi-squared value for the crystal structures(s) using CRY SOL?

ANSWER: The chi-squared value using CRY SOL is shown in Supplementary Table 2.

19. line 124: what program produced the Z-score? Was it DALI? If so, it should be mentioned and cited.

ANSWER: We agree with the reviewer. Z-score values were produced by using DALI. We have included a reference accordingly.

Holm, L., Rosenström, P. Dali server: conservation mapping in 3D. Nucl. Acids Res. 38, W545-549 (2010).

20. Fig. 2: The choice of bright, reflective colours for the molecular surface makes it very hard to see the bound product. It's not clear what the point of panel b) is.

ANSWER: We agree with the reviewer. We have modified the Figure accordingly.

21. Online methods:

line 262: the reference 21 should be moved to just after "FastCloning method" on line 261 and an address should be given for GeneWiz.

ANSWER: We have modified the text accordingly.

22. line 265: I guess 50 micrograms/ml is meant

ANSWER: We have modified the text accordingly.

23. line 275: Coomassie Blue

ANSWER: We have modified the text accordingly.

24. line 279: What happened after the protein was loaded onto the Superdex column? I guess it was eluted in some way? How was it concentrated and stored?

ANSWER: We agree with the reviewer. In the new version of the manuscript we have extended the experimental details on the protein purification protocol, as requested.

25. line 283: sodium

ANSWER: We have modified the text accordingly.

26. line 284: amino acid mixture (L-Na-glutamate,... lysine HCl)

ANSWER: We have modified the text accordingly.

27. line 290: How were the data scaled, with XSCALE or AIMLESS in CCP4?

ANSWER: The data was scaled by using XSCALE, the stand-alone scaling program of the XDS suite.

28. line 294: Was 4NUZ used unmodified as a search model, or was it split into domains?

ANSWER: We have used the unmodified 4NUZ pdb entrance as a search model. We have included this information in the text.

29. line 301: SAXS is not diffraction.

ANSWER: We agree with the reviewer. We have modified the text accordingly.

30. line 303: batch mode

ANSWER: We have modified the text accordingly.

31. line 305: The description of SAXS data collection is unsatisfactory. What was the sample size?

ANSWER: We have extended the description of SAXS data collection as suggested. The sample size was 50 μ L.

32. Capillary volume?

ANSWER: The capillary volume of the sample was 30 μ L (1.5 mm diameter capillary with 10 μ m wall thickness). We have modified the text accordingly.

33. Temperature?

ANSWER: We have performed the SAXS experiments at 25°C. We have included this information in the Table S2.

34. Was the sample flowed or oscillated to avoid radiation damage?

ANSWER: The sample flow was extremely low (High 10^{10} photons/second). In addition, we have used 50 mM Tris and 2 % glycerol in the buffer to mitigate radiation damage as well.

35. Was the I(0) normalised?

ANSWER: The I(0) was normalized considering protein concentration.

36. line 311: Were any attempts made to do ab initio reconstruction using DAMMIN/DAMMIF?

ANSWER: We have performed *ab initio* reconstructions by using DAMMIN/DAMMIF. We obtained similar results than with GASBOR.

37. line 319: The lack of conformational change is a result and should not be mentioned only in the Online Methods section, but in the main text.

ANSWER: We agree with the reviewer. We have incorporated this information in the new 'Results and Discussion' section.

38. The supplementary/supplemental/supporting material (the authors use all three terms) is a bit sloppily prepared, with above all inadequate figure legends.

ANSWER: We have made a strong effort to improve the readability of the Supplementary Section.

39. Table S1: The table seems to have been copied from phenix.table_one without too much afterthought. The space group is more simply written P2(1), with the 1 as a subscript.

ANSWER: We have modified the Supplementary Table 1 accordingly.

40. The use of two decimal places for unit cell dimensions and for the B-factors is not necessary.

ANSWER: We have modified the Supplementary Table 1 accordingly.

41. The difference between Rmodel and Rfree seems rather small for a 2.9 Å structure: were the Rfree flags used to cross-validate the unbound structure transferred to the dataset for the bound structure presented here? It isn't stated anywhere how many reflections were used for the Rfree calculation.

ANSWER: The difference between the Rmodel and Rfree is of 3% because we have used weight optimization in the final round of refinement. The Rfree flags were not transferred between datasets. We have included the number of reflections used for the Rfree calculation on the Table S1.

42. Table S2: The symbol \leq should presumably be \pm . There seems to be an extra digit, "1" after the end of the second Porod volume estimate.

ANSWER: We have modified the Supplementary Table 2 accordingly.

43. The units of $I(0)$ cannot be reciprocal Ångström, as this is the unit of q , on the x-axis of the scattering curve. $I(0)$ is on the y axis and has arbitrary units unless normalised using a known standard (was this done?) Likewise, why is the unit reciprocal cm used for the $I(0)$ estimate from Guinier?

ANSWER: We agree with the reviewer. In the new version of the manuscript, we have modified the units of $I(0)$ in the Supplementary Table 2 and the y axis in Supplementary Figure 4a. We have used ScÅter to calculate the molecular weight. Please see reference:

Rambo, R. P., and Tainer, J. A. Accurate assessment of mass, models and resolution by small-angle scattering. *Nature*. 496, 477–481 (2013).

44. "Comparation" should be "comparison".

ANSWER: We have modified the text accordingly.

45. Why is the CRY SOL chi-squared value so high? Did the authors consider EOM?

ANSWER: We interpret that the chi-squared is high because the EndoS shows some flexibility in solution as depicted in the normalized Kratky plot (please see new Figure S4b). Following the reviewer's suggestion, we have used EOM and SREFLEX to evaluate the flexibility of the system. We have obtained the best chi-squared values using SREFLEX and we have included this new data in the new version of the manuscript.

46. The program "SCATER" should be spelled "SCÅTTER"

ANSWER: We have modified the text accordingly.

47. In Fig. S1, the figure gives the misleading impression that the glycosylations are highly exposed on the exterior of the Fc domain, when they are in fact on the inside. While I appreciate that this can be hard to draw, it should at least be indicated in the figure legend. It should be explained what the right-hand reactive group of the molecule used to attach uniform glycosylations to IgG is. The cleavage point leading to the G2 product should be indicated. The symbols for the different types of saccharide unit should be defined. There should be a legend for this figure!

ANSWER: We have modified the Supplementary Figure 1 (now Figure 1) and its legend accordingly.

48. Fig. S2 legend: "996-1191 residues" should read "residues 996-1191". The sequence should be numbered.

ANSWER: We have modified the text accordingly.

49. In Fig. S4, the same colour scheme should be used for the ab initio reconstructions as for the scattering curves and P(r) functions. As it is, the orange colour is used for the complexed protein in panel a and for the uncomplexed protein in panel b.

ANSWER: We have modified the Supplementary Figure 4 accordingly.

50. In Fig. S6, the legend states that the complexed structure is orange in both panels a and b, when it is only orange in panel A. What colour scheme is used for panel b? Is the uncomplexed structure still light grey?

ANSWER: We have modified the Supplementary Figure 6 (now Supplementary Figure 3). Please see Supplementary Figure 4c.

51. In Fig. S7, the structure-based sequence alignment cannot be completely correct, because e.g. the second of the catalytic residues marked by red dots is not conserved in EndoF1 and EndoBT.

ANSWER: We have modified the corresponding Supplementary Figure (now Supplementary Figure 5), accordingly.

52. Fig. S8 seems to show a biochemical synthesis rather than a chemical one.

ANSWER: We have modified the text accordingly. Please see Supplementary Figure 6.

Reviewer #2:

1. This manuscript describes the structure of a complex between the IgG-specific endoglycosidase, EndoS, and a biantennary complex type N-glycan. The resulting structure describes the manner by which the endoglycosidase recognizes the branched glycan product in the enzyme active site. While the manuscript is an effective structure presentation of the enzyme-glycan product complex, there are some deficiencies that need to be resolved in the description of the complex, including:

The authors completely under-represent the fact that they already published the overall structure of apo-EndoS (reference 18, Trastoy et al (2014) PNAS). The reference of this prior paper was cryptically cited during the structure description. However, it is so buried that a cursory read of the present paper would have left a reader with an impression that this was the first structural description of this protein. As such, the novelty of the present paper is the structure of the glycan substrate complex, not the initial description of the protein. The details of the complex are well

described in the manuscript and the mutagenesis data, coupled with kinetic analysis, are an effective description of how the recognition of the biantennary complex glycan structure is achieved.

ANSWER: We very much appreciate your considerations about the novelty of our study, and all the constructive comments/suggestions you made to improve the article.

The original manuscript was transferred from another NPG journal, with a considerably more condensed format. In the process to reshape the manuscript, the comments made by the reviewers have been invaluable. In that sense, (i) we have included the original unliganded crystal structure from reference 18 in the new ‘Introduction’ and ‘Results and Discussion’ sections and (ii) extended the comparison between both crystal structures in the new ‘Results and Discussion’ section and new Supplementary Figures 3 and 4.

2. In Fig 3C the authors attempt to make a structural comparison with other endoglycosidase structures, but this visual representation is not effective. The nature of the isolated loop regions in cartoon representation in the illustration provide no meaningful insight to the differences in structural basis of substrate specificity. The authors state that “Residues involved in the interaction with the conserved Man β 1–4GlcNAc β 1–4GlcNAc β 1– core are essentially preserved ...” This conservation in interactions with these glycan residues are not evident from this figure.

ANSWER: In this new version of the manuscript, we have described in detail the structural differences between members of the GH18 family of endoglycosidase to support the substrate specificity. Please see ‘Structural comparison of EndoS with GH18 family of endoglycosidases’ in the ‘Results and Discussion’ section and the new Figures 8 and 9. The previous Figure 3c (now Figure 7), is complemented by the structure base alignment reported in Supplementary Figure 5.

What are the PDB files used in this comparison along with the respective citations of the structures?

ANSWER: We have included the PDB codes and the respective citations of the structures in the main text.

How many of the structures have substrate complexes?

ANSWER: Only one. EndoF₃ in complex with the G2 product.

The authors make cursory statements about the comparison with the substrate complex for EndoF₃. The previous PNAS paper presented a structural comparison of EndoS with the EndoF₃-glycan complex, but the present manuscript needs to show a detailed structural comparison of the EndoF₃ complex with the EndoS-glycan complex. What is novel about the present glycan structural complex by comparison to the prior work on EndoF₃?

ANSWER: In the new version of the manuscript, we have now included an extensive and detailed comparison and discussion on the structural differences of the EndoS and EndoF₃ glycosidase domains, and the importance of these differences to define the glycan specificity of

each enzyme. Please see ‘Structural comparison of EndoS with GH18 family of endoglycosidases’ in the new ‘Results and Discussion’ section, and new Figures 8 and 9.

3. In the prior PNAS paper the authors present a model of the docked EndoS complex with IgG and how the EndoS accessory domains enable the insertion of the glycan into the active site. Is this model still relevant based on the present glycan complex?

ANSWER: Although we expect the IgG to accommodate into the concave interior of EndoS (please see Figure S6 in PNAS paper), the new EndoS-G2 crystal structure strongly support the occurrence of an important conformational change on the Fc fragment (i) to expose the glycan/s and (ii) to accommodate them into the deep grooves of the glycoside hydrolase domain.

4. The authors model in the GlcNAc(-1) and NeuAc(+6) and (+10) monosaccharides in Fig. S5, but never describe how that modeling was performed nor is it described in either the legend to Fig. S5 or the body of the paper that these monosaccharides were modeled and not a part of the empirically derived structure.

ANSWER: We agree with the reviewer. We have included this information in the Methods section.

5. It would be very helpful to show the positions of the catalytic residues that were mutated and provide a model for the catalytic mechanism that resulted in the formation of the enzymatic product complex that is present in the active site. This would best be presented in conjunction with Fig. 2D.

ANSWER: We agree with the reviewer. As suggested, we have included a new Figure 5 displaying (i) the proposed catalytic mechanism of the reaction and (ii) the position of the mutated residues in the active site.

6. In regard to the N-3HB domain, the authors state that: “these data suggest that the N-3HB domain is a key element involved not only in N-linked glycan substrate recognition but also in EndoS-Fc fragment interaction.” The data do not really suggest these conclusions. An equally likely interpretation is that the N-3HB domain stabilizes the GH domain structure. The IgG domain would be expected to engage the active site from the opposite side of the GH domain from the N-3HB domain and would not likely directly interact with the IgG domain.

ANSWER: We agree with the reviewer. We have performed differential scanning fluorimetry (DSF) in order to study the thermostability of full length EndoS and that of Δ N-3HB-EndoS. We observe that Δ N-3HB-EndoS shows two well-separated unfolding transition states with a melting temperature (T_m) of 51 °C and 45 °C, whereas full length EndoS only presents one transition at the corresponding higher T_m value (please see Supplementary Figure 3f). As suggested by the reviewer, the N-3HB domain could be involved in the stabilization of the glycoside hydrolase domain. We have included/discussed this experimental data in the new ‘Results and Discussion’ section of the manuscript.

7. Other minor issues:

Please move the Man+7 label out into white space in Fig 2D.

ANSWER: We have modified the Figure accordingly (now Figure 3d).

8. Show residue D233A in Fig 2D.

ANSWER: We have modified the Figure accordingly (now Figure 3d).

9. Overall, the manuscript is a nice structural presentation of the enzyme-glycan product complex for EndoS and provides tremendous insights into how this enzyme is able to recognize biantennary complex glycan structures. Additional detail is required to provide insights into the novelty of this structure in compared to the prior PNAS paper and to the structure of the EndoF3-glycan complex. No additional experimentation is required, but a more effective presentation of comparative data with other structures would greatly improve the manuscript.

ANSWER: We very much appreciate that you consider our study provides a tremendous insight into how this enzyme is able to recognize biantennary complex glycan structures, and all the constructive comments/suggestions to improve the article. Following the reviewer suggestions, we have made a strong effort in order to effectively present the comparison of the structural data and improve the readability of the entire manuscript.

Reviewer #3:

1. IgG glycans regulate antibody effector functions. EndoS from *Streptococcus pyogenes* cleaves specifically complex-type N-linked glycans of IgG and contributes to the immune evasion of the bacterium. Interestingly, EndoS glycosynthase variants can be used to transfer pre-defined complex type N-glycans to intact IgG, which could be very useful to generate therapeutic antibodies with the desired glycoforms. With the aim to better understand the substrate specificity of this enzyme, Trastoy and collaborators have crystallized an inactive variant of endoS in complex with its oligosaccharide product (G2 glycoform). Following the identification of EndoS domains interacting with the G2 glycosform, authors have performed key mutations of amino acid residues to demonstrate their involvement in the recognition of the glycan product. Finally, by comparing the 3D structure of EndoS domains with those of other glycosylhydrolases members of the GH18 family, authors proposed that the substrate specificity of EndoS relies on the unique groove 2 of the enzyme. Although the manuscript is clearly written, of great interest and provides several novel findings, additional experiments are required to fully support the conclusions made by authors:

ANSWER: We very much appreciate that you consider our study to be of great interest, and all the constructive comments/suggestions you made to improve the article.

2. Major comments:

My main concern is related to the last part of the manuscript. Based on the functional data depicted in figure 3 and comparison of endoglycosidase 3D structures, authors proposed that the EndoS specificity for complex-type N-linked oligosaccharide relies on its exclusive groove 2 structure (loops 1, 2 and 7), which is significantly different than those of other GH18 endoglycosidases. But authors did not directly demonstrate it. Authors mentioned (Lines 141-143 and Fig. 3C) that loops 1 and 2 of high-mannose-type specific GH18 endoglycosidases are “ordered” and adopt “a beta-hairpin conformation” but not that of endoS. However, a counter example of this is the lack of order of loops 1 and 2 of EndoT, which is also a high-mannose-type specific endoglycosidase.

We have modified Figure 3c (new Figure 7) to show the correct loop 2 of EndoT and confirm that it is also a β -hairpin like in the other high-mannose-type specific endoglycosidases.

3. Likewise, (Line 144), authors mentioned that the Loop7 is “markedly shorter” in high-mannose-type specific GH18 endoglycosidases as compared to that of EndoS. However, EndoF3 that recognizes complex-type N-glycans, also have a very short loop 7.

EndoF₃ has a shorter loop 7 than EndoS because EndoF₃ can accept both triantennary and biantennary complex type oligosaccharides as substrates (please see Figure 8). EndoS is not able to hydrolyze triantennary complex type oligosaccharides. As depicted in Figure 8, the long loop 7 observed in EndoS would block the entrance of the third antenna.

4. A direct and more convincing proof of the involvement of loop 1, 2 and 7 roles in the substrate specificity of EndoS would be the substitution or introduction of residues (or even entire loops) from corresponding loops in high-mannose-type-specific GH18 endoglycosidases (Ex. EndoH), following by the measurement of binding of different glycan products (G2, High-mannose-type glycans,...) or, if possible, measurement of activity of EndoS variants on different substrates.

ANSWER: Switching loops between endoglycosidases might be a challenging task in order to keep a correct folding of the conserved $(\alpha/\beta)_8$ barrel of the glycoside hydrolase domains. The proposed study is out of the scope of the current manuscript. However, following the reviewer suggestion, we provide additional support on the correlation between the architecture of the endoglycosidase loops surrounding the glycan binding pocket and the specific activity of this family of enzymes. Taking into account the analysis performed in Figure 7, the architecture of loops 1, 2 and 7 in EndoBT is similar to that observed in EndoT, EndoH and EndoF₁, strongly suggesting that the enzyme is an endoglycosidase specific for high-mannose type oligosaccharides. It is worth mentioning that the crystal structure of EndoBT, a putative endoglycosidase of unknown function/enzymatic activity/glycan specificity, was solved in its unliganded form. We therefore determined the capacity of EndoBT to hydrolyse biantennary complex-type N-linked glycans and/or high mannose type N-linked glycans from IgG1 antibodies. As depicted in Figure 9e, the hydrolytic assays showed that EndoBT is able to hydrolyse high-mannose type IgG1 but not biantennary complex-type IgG1.

3. Minor comments

Figures 3a and b- mutations are presented in terms of amino acids (ex. R119A/E130A/K133A) whereas the corresponding text (Lines 104-128) refers to Loops, making the demonstration hard to follow. Please clearly present the Loops also in both figures.

ANSWER: We have modified the corresponding Figures 3a and 3b (new Figures 6a and 6b) accordingly.

4. Figure 3c: For Endo S, Loop-1 is indicated to interact with two of the core mannoses whereas Figure 3a suggest that Loop-1 interacts with terminal glucosamine and galactose. Please, correct or explain this discrepancy.

ANSWER: The reason is the orientation of the G2 product in the Figure. Arg119 interacts with Man (-7) whereas Trp121 interacts with Man (-7) and Man (-2). The terminal GlcNAc (-10) and Gal (-9) residues of the G2 product protrude away from groove 2 of EndoS.

5. Figure 3c: Loops 3, 4, 5 and 6 are not at all commented into the text, compromising the interest of presenting them into the table. I would suggest moving them in supplementary data

ANSWER: We thank the reviewer for the suggestion. However, we prefer to present all the data in the Figure.

6. Figure 3b- A presentation of mutant ΔN is lacking in the legend.

ANSWER: We agree with the reviewer. We have modified the text accordingly.

7. Line 53- typo « glycoformsand and »

ANSWER: We have modified the text accordingly.

8. Line 69- define or explain « hybrid Ig domain »

ANSWER: We agree with the reviewer. We have included a description on the hybrid Ig domain. This domain is composed of two subdomains that are topologically entwined. The smaller of the two subdomains is inserted within the loop that connects the second and third β -strands of the larger subdomain, which is a typical Ig domain structurally similar to the interleukin-4 receptor (PDB ID code 1IAR; Z score = 5.2).

9. Line 95- Correct: “residues of each arm adopt two “alternative” (instead of different) conformations” and precise “into the crystal”. Likewise, in Figure S5- Correct “the two alternative conformations of G2 product”.

ANSWER: We have modified the text accordingly.

10. Line 125- the mode of interaction of the SpA C domain (similar to endoF N-3HB) with IgG or IgM is not clearly explained. The reader cannot even discriminate between protein-protein or protein-glycan interfaces ...

ANSWER: We agree with the reviewer. In the new version of the manuscript, we have made an effort to clearly explained the proposed mode of interaction of the SpA C domain with IgG or IgM and added references accordingly.

11. Figure S1- For a better understanding, please add some explanations on the fig S1 legend. I also don't understand why the item "Immune evasion" is below the second arrow and not below the first arrow.

ANSWER: We have modified the Figure S1 (new Figure 1) and the associated legend.

12. Figure S6- Line 110- Correct "a" and "b" instead of "A" and "B".

ANSWER: We have modified the Figure accordingly (please see new Supplementary Figure 3).

13. Figure S6A- Loop 2 should be highlighted in the figure.

ANSWER: We have modified the Figure S6a (new Figure 3c,d) accordingly.

14. Lines 138-139- steric hindrance is hypothesized for EndoS, and compared to tridimensional permissiveness in EndoF3 or EndoH, but no 3D illustration of EndoF3 or H groove 2 corroborate the comments (as Figures 3c focuses on EndoS and Figure S7 presents truncated loops in a series of Endo enzymes). This should be illustrated.

ANSWER: We agree with the reviewer. We have added two new figures illustrating the EndoS, EndoF₃, EndoH, EndoF₁, EndoT and EndoBT surfaces (Figures 8 and 9). We have also included new molecular docking experiments of high mannose-type oligosaccharide GlcNAc₁Man₉ into the GH18 family endoglycosidases able to hydrolyze high mannose-type N-linked glycans and tri-antennary complex-type oligosaccharide into EndoF₃. In these figures we can observed that the binding pocket of EndoS is unique in accommodating bianntenary complex-type N-linked glycans while excluding other glycan types due to steric hindrance.

We wish to take this opportunity to thank the reviewers for her/his thoughtful suggestions that have made the manuscript so much better.

Reviewers' comments:

Reviewer #1 (Remarks to the Author):

The revised manuscript by Trastoy et al. is a significant improvement on the original version. The authors have answered all of my original questions and I think that the article is now almost suitable for publication. The detailed comparison with EndoF3 is much appreciated. However there are still some errors in the manuscript that need to be corrected.

The first time that Z-scores are mentioned it should read "DALI Z-score" and a reference to DALI should also be inserted there.

On page 5, line 18 "the calculated buried surface area ... was" should be "is", as it presumably hasn't changed since it was calculated.

Same page, line 19: "form hydrogen bonds"

On p.8 line 11 there is a mysterious bold text "BEA", which seems to be a request for the first author to complete the sentence. However this hasn't been done.

p. 8: It's still unclear to me what evidence the current structure provides for the proposed reaction mechanism for GH18 enzymes apart from the fact that important residues are present. Does the product conformation say anything about the proposed distortion of the substrate?

The loop containing residues 303 and 305 is referred to as loop 6 in the text but loop 5 in Fig. 6. Also in Fig. 6, loop 3 is described as being light red when in fact it is some kind of gold colour. Please check all nomenclature and colours so that everything is consistent.

p. 10: How can the cavity be bigger in EndoS (almost 3 times the size that it is in EndoF3) when EndoF3 has the ability to bind a triantennary glycosylation? Are the authors comparing like with like?

p. 11: What do the authors mean by an "exclusive" groove 2? Groove 2 is present in both EndoF3 and EndoS, since both can bind the G2 product. EndoS excludes the triantennary substrate by sterically blocking a groove that is present in EndoF3.

One technical point that I missed the first time around:

In the SAXS figures and tables, it is not clear whether the curves and values obtained were obtained from a single concentration or through the merging of data from the several concentrations used. This is an important detail, and it should be included in the Methods and figure and table legends.

One last point that is more a question of taste:

pdb should be written PDB, as it is an abbreviation for Protein Data Bank.

Reviewer #2 (Remarks to the Author):

This revised manuscript describes the structure of a complex between EndoS, and a biantennary complex type N-glycan substrate analog to map the structural determinants of substrate specificity. The revised manuscript was greatly improved in response to prior reviewer critiques, but still has several concerns that should be addressed prior to publication.

- 1) There are numerous places in the revised manuscript where the text description essentially catalogs structural information that would be better handled as a presentation in table form. Examples include manuscript lines 124-127, lines 131-142, lines 198-205, but similar issues are present in other parts of the manuscript. This catalog-like description breaks up the readability of the manuscript and the level of detail is not essential for the body of the manuscript text.
- 2) While the manuscript now mentions the structure determination of EndoS in their prior PNAS paper, the present paper has an extended description of the overall structure of the multi-domain protein (lines 83-91) that has already been described in the prior paper. A more appropriate presentation would to a focus on the differences in the present structure relative to the prior published structure and point readers to the prior paper for a description of the overall domain structure.
- 3) The present studies on full length EndoS employ a double mutant for (D233A/E235L) of the recombinant enzyme, but the authors never state why they chose these residues to mutate. These are not the mutants employed in the prior work and it is unclear why they needed to employ a mutant form of the enzyme at all, given the structure of the ligand resembling the enzymatic product in the structural studies. The authors should explain the basis for these mutations. There is an oblique comment on line 172 regarding the fact that the double mutant is inactive, but no description why these residues were employed.
- 4) Throughout the manuscript the authors state that they are employing a G2 structure as the substrate analog, but in reality the ligand is a truncated structure that represents the equivalent of the enzymatic product of EndoS action (cleaved between the two core GlcNAc residues). This should be clarified in Fig 5A. Again, it is not clear why the D233A/E235L mutant was employed and how these mutations impacted in the interaction with the reducing terminal GlcNAc residue.
- 5) On line 161 the authors should define BEA.
- 6) Line 181 states that mutations in loop 6 abolished hydrolytic activity and the data is shown in Fig. 6 A. Loop 6 mutants are not shown in Fig. 6A or any other figure.
- 7) The structural comparisons illustrated in Fig. 8, Fig. 9 and supplementary S5 are effective in examining substrate interactions in the respective enzyme catalytic domains. However, it would also be helpful to compare the overall structures of the respective proteins. Do the other enzymes have the equivalent of the additional accessory domains found in EndoS? If so, then how to they compare? If not, that is also relevant, because it implies that the additional domains in EndoS are employed for selective interactions with Igs that are not present nor relevant for the other enzymes.

8) It would be beneficial to have some sort of overlay figure, at least for EndoS and EndoF3, showing the similar positions of catalytic residues as described in manuscript lines 217-220.

9) Figure 5A: What is the meaning of the regions labeled 'R1' and 'R2'? This would be an appropriate place to indicate that the G2 ligand bound in the EndoS active site has a single GlcNAc at its reducing terminus and represents the enzymatic product.

Overall, the manuscript is greatly improved and presents a nice structural presentation of the enzyme-glycan product complex for EndoS and provides tremendous insights into how this enzyme is able to recognize biantennary complex glycan structures. Revisions to address the minor points raised above are suggested.

Reviewer #1:

The revised manuscript by Trastoy et al. is a significant improvement on the original version. The authors have answered all of my original questions and I think that the article is now almost suitable for publication. The detailed comparison with EndoF3 is much appreciated. However, there are still some errors in the manuscript that need to be corrected.

1. The first time that Z-scores are mentioned it should read "DALI Z-score" and a reference to DALI should also be inserted there.

ANSWER: We agree with the reviewer. We have modified the text accordingly.

2. On page 5, line 18 "the calculated buried surface area ... was" should be "is", as it presumably hasn't changed since it was calculated.

ANSWER: We agree with the reviewer. We have modified the text accordingly.

3. Same page, line 19: "form hydrogen bonds"

ANSWER: We have modified the text accordingly.

4. On p.8 line 11 there is a mysterious bold text "BEA", which seems to be a request for the first author to complete the sentence. However, this hasn't been done.

ANSWER: We agree with the reviewer. We have removed the text accordingly.

5. p. 8: It's still unclear to me what evidence the current structure provides for the proposed reaction mechanism for GH18 enzymes apart from the fact that important residues are present. Does the product conformation say anything about the proposed distortion of the substrate?

ANSWER: We did not observe a distortion of the GlcNAc (-1) moiety.

6. The loop containing residues 303 and 305 is referred to as loop 6 in the text but loop 5 in Fig. 6. Also in Fig. 6, loop 3 is described as being light red when in fact it is some kind of gold colour. Please check all nomenclature and colours so that everything is consistent.

ANSWER: We have modified the text accordingly.

7. p. 10: How can the cavity be bigger in EndoS (almost 3 times the size that it is in EndoF3) when EndoF3 has the ability to bind a triantennary glycosylation? Are the authors comparing like with like?

ANSWER: EndoS and EndoF₃ cavities were compared using the same parameters. EndoF₃ cavity is much more solvent exposed compared to that observed in EndoS.

8. p. 11: What do the authors mean by an "exclusive" groove 2? Groove 2 is present in both EndoF₃ and EndoS, since both can bind the G2 product. EndoS excludes the triantennary substrate by sterically blocking a groove that is present in EndoF₃.

ANSWER: We have removed the word 'exclusive' for clarity.

9. One technical point that I missed the first time around: In the SAXS figures and tables, it is not clear whether the curves and values obtained were obtained from a single concentration or through the merging of data from the several concentrations used. This is an important detail, and it should be included in the Methods and figure and table legends.

ANSWER: We agree with the reviewer. We have obtained the data through merging data from several concentrations of protein and ligand. We have included this information in the Methods sections.

10. One last point that is more a question of taste: pdb should be written PDB, as it is an abbreviation for Protein Data Bank.

ANSWER: We agree with the reviewer. We have modified the text accordingly.

Reviewer #2:

This revised manuscript describes the structure of a complex between EndoS, and a biantennary complex type N-glycan substrate analog to map the structural determinants of substrate specificity. The revised manuscript was greatly improved in response to prior reviewer critiques, but still has several concerns that should be addressed prior to publication.

1. There are numerous places in the revised manuscript where the text description essentially catalogs structural information that would be better handled as a presentation in table form. Examples include manuscript lines 124-127, lines 131-142, lines 198-205, but similar issues are present in other parts of the manuscript. This catalog-like description breaks up the readability of the manuscript and the level of detail is not essential for the body of the manuscript text.

ANSWER: We very much appreciate the suggestion of the reviewer but we prefer describe/incorporate the structural information in the main text.

2. While the manuscript now mentions the structure determination of EndoS in their prior PNAS paper, the present paper has an extended description of the overall structure of the multi-domain protein (lines 83-91) that has already been described in the prior paper. A more appropriate presentation would to a focus on the differences in the present structure relative to the prior

published structure and point readers to the prior paper for a description of the overall domain structure.

ANSWER: We have modified the text accordingly.

3. The present studies on full length EndoS employ a double mutant for (D233A/E235L) of the recombinant enzyme, but the authors never state why they chose these residues to mutate. These are not the mutants employed in the prior work and it is unclear why they needed to employ a mutant form of the enzyme at all, given the structure of the ligand resembling the enzymatic product in the structural studies. The authors should explain the basis for these mutations. There is an oblique comment on line 172 regarding the fact that the double mutant is inactive, but no description why these residues were employed.

ANSWER: E235 acts as a general acid/base, whereas D233 stabilizes the intermediate in a substrate-assisted hydrolysis mechanism in which the carbonyl group of the C2-acetamido of GlcNAc (-1) acts as the nucleophile (Fig. 5b). For that reason, we replaced both D233 and E235 residues by alanine and leucine, respectively, in order to obtain a catalytically inactive enzyme (EndoSD233A/E235L) for further use in our structural studies.

4. Throughout the manuscript the authors state that they are employing a G2 structure as the substrate analog, but in reality the ligand is a truncated structure that represents the equivalent of the enzymatic product of EndoS action (cleaved between the two core GlcNAc residues). This should be clarified in Fig 5A. Again, it is not clear why the D233A/E235L mutant was employed and how these mutations impacted in the interaction with the reducing terminal GlcNAc residue.

ANSWER: We have modified the text and Figure 5a accordingly. Please see also point 3.

5. On line 161 the authors should define BEA.

ANSWER: We have removed the text accordingly.

6. Line 181 states that mutations in loop 6 abolished hydrolytic activity and the data is shown in Fig. 6 A. Loop 6 mutants are not shown in Fig. 6A or any other figure.

ANSWER: We have removed the text accordingly.

7. The structural comparisons illustrated in Fig. 8, Fig. 9 and supplementary S5 are effective in examining substrate interactions in the respective enzyme catalytic domains. However, it would also be helpful to compare the overall structures of the respective proteins. Do the other enzymes have the equivalent of the additional accessory domains found in EndoS? If so, then how do they compare? If not, that is also relevant, because it implies that the additional domains in EndoS are employed for selective interactions with Igs that are not present nor relevant for the other enzymes.

ANSWER: We have included a new Supplementary Fig. 6 that shows the full length structures of all the endoglycosidases discussed in this study. We have also mentioned in the main text that EndoBT shows an additional carbohydrate binding module (CBM) domain.

8. It would be beneficial to have some sort of overlay figure, at least for EndoS and EndoF3, showing the similar positions of catalytic residues as described in manuscript lines 217-220.

ANSWER: We agree with the reviewer. In the new version of the manuscript we have incorporated this information in Figure 5b.

9. Figure 5A: What is the meaning of the regions labeled 'R1' and 'R2'? This would be an appropriate place to indicate that the G2 ligand bound in the EndoS active site has a single GlcNAc at its reducing terminus and represents the enzymatic product.

ANSWER: The regions R1 and R2 are substituents of O-4 and O-1 of the GlcNAc(-1), respectively. This is showed in Figure 5a.

Overall, the manuscript is greatly improved and presents a nice structural presentation of the enzyme-glycan product complex for EndoS and provides tremendous insights into how this enzyme is able to recognize biantennary complex glycan structures. Revisions to address the minor points raised above are suggested.

ANSWER: We wish to take this opportunity to thank the reviewers for her/his thoughtful suggestions that have made the manuscript so much better.